# Neural correlates of hierarchical predictive processes in autistic adults

Laurie-Anne Sapey-Triomphe [1,2] ✉, Lauren Pattyn[1], Veith Weilnhammer[3,4], Philipp Sterzer [3,4] & Johan Wagemans [1,2]

Bayesian theories of autism spectrum disorders (ASD) suggest that atypical predictive mechanisms could underlie the autistic symptomatology, but little is known about their neural correlates. Twenty-six neurotypical (NT) and 26 autistic adults participated in an fMRI study where they performed an associative learning task in a volatile environment. By inverting a model of perceptual inference, we characterized the neural correlates of hierarchically structured predictions and prediction errors in ASD. Behaviorally, the predictive abilities of autistic adults were intact. Neurally, predictions were encoded hierarchically in both NT and ASD participants and biased their percepts. High-level predictions were following activity levels in a set of regions more closely in ASD than NT. Prediction errors yielded activation in shared regions in NT and ASD, but group differences were found in the anterior cingulate cortex and putamen. This study sheds light on the neural specificities of ASD that might underlie atypical predictive processing.

Even though we live in a complex and changing environment, most of us manage to minimize uncertainty and therefore avoid being overwhelmed by surrounding stimuli. This is achieved using our predictive abilities which rely on prior beliefs. Priors are accumulated over previous experiences and capture the hierarchical statistical regularities of sensory stimuli. Using a generative model, the brain would act like a predictive engine that constantly tries to anticipate or predict the incoming inputs, and infer their causes[1–3]. This process has been formalized in the Bayesian brain framework, where the brain implements and updates a hierarchical model of the world[3–5]. Priors are combined with sensory inputs to generate percepts, and their relative weights depend on their precision (i.e., inverse variance). The prior/sensory precision balance needs to be flexibly adjusted to the context to yield optimal perception. Discrepancies between priors and sensory inputs are signaled through prediction errors, which can be used to update priors or may be ignored when signaling irrelevant noise[6]. Predictive mechanisms are hypothesized to rely on a bidirectional cascade of top-down predictions and bottom-up prediction errors, across hierarchically organized cortical layers and brain regions[7].

Impaired predictive skills would lead to an atypical perception and a sensation of an unpredictable world. In particular, atypical predictive mechanisms might underlie the symptoms encountered in Autism Spectrum Disorders (ASD)[8–14]. ASD is characterized by deficits in social interactions and communication, and by restrictive and repetitive behaviors and interests (Diagnostic and Statistical Manual of Mental Disorders, 5th edition, DSM-5[15]). Even though this neurodevelopmental condition affects >1% of the population[16], the core mechanisms of ASD remain to be identified. The Bayesian theories of ASD offer a promising framework to account for the heterogeneous symptoms of ASD. Indeed, suboptimal predictive abilities might, for instance, explain the higher uncertainty intolerance in autistic individuals[17] or their difficulties to rapidly build up an internal representation to categorize stimuli[18]. More broadly, if ASD is a disorder of perceptual inference[19], it would prevent autistic individuals from accurately inferring the meaning of stimuli or making predictions. The social domain would be particularly impacted as it is a complex, dynamic, and noisy domain, where predictions need to be frequently updated and adjusted to the context. Non-social symptoms, such as

[1]Department of Brain and Cognition, Leuven Brain Institute, KU Leuven 3000 Leuven, Belgium. [2]Leuven Autism Research (LAuRes), KU Leuven 3000 Leuven, Belgium. [3]Department of Psychiatry, Charité-Universitätsmedizin Berlin, 10117 Berlin, Germany. [4]Berlin Institute of Health, Charité-Universitätsmedizin Berlin, 10178 Berlin, Germany. ✉e-mail: laurie-anne.sapey-triomphe@inserm.fr

repetitive behaviors or insistence on sameness, could be a way to restore some predictability. More specifically, Bayesian theories have suggested a reduced impact of priors in ASD, either because of low prior precision[11] or because of high sensory precision[8], both resulting on a very realistic perception of the world according to these theories. One of the hypotheses focusing on the ratio of prior and sensory precisions[9,10,13] is that there might be a high and inflexible precision of prediction errors in autism (HIPPEA)[13].

As detailed in a recent review[20], the predictive abilities of autistic individuals are intact or impaired depending on the context. While the perceptual bias induced by structural priors seems typical in ASD (e.g., ref. 21,22), the learning of predictive associations is sometimes impaired[20]. Associative learning would be particularly difficult in ASD when predictive features have low salience or consistency[20]. In terms of learning dynamic, autistic individuals were slower at updating predictions[23], which could explain a more inflexible weighting of priors in ASD[24]. The ability to learn probabilistic associations in volatile environments was intact in autistic children[25], but atypical in autistic adults[26] who showed less surprise to unexpected outcomes in ASD[26], consistently with the hypothesis of more inflexible prediction errors in ASD[13]. This is also in line with a mismatch negativity study showing a less flexible modulation of prediction errors in autistic adults[27].

Despite the thought-provoking debate about the Bayesian hypotheses of ASD, very little is known about their neural correlates in ASD. Three fMRI studies recently investigated reward and social prediction errors in ASD. In a false belief paradigm, social prediction errors were atypically encoded in the gyral surface of the anterior cingulate cortex (ACC) of autistic adults[28]. In two studies investigating the neural correlates of reward prediction errors, group differences between neurotypicals (NT) and autistic individuals were found in the paracingulate gyrus, insula, and frontal pole[29] and in the ACC and frontal regions[30]. Yet, note that these two last studies did not identify neural correlates of prediction errors within groups. The neural mechanisms underlying predictions and prediction errors in ASD should be characterized in order to shed light on the predictive processes in ASD.

Unlike the literature on ASD, the neural correlates of predictive mechanisms have been quite extensively investigated in NT[31–36]. Predictions and prediction errors are thought to pass between superficial and deeper cortical layers within each brain region (e.g., ref. 37), and some key regions have been identified. For instance, when processing contextual associations, the parahippocampal and retrosplenial cortices represent familiar associations at different levels of abstraction, while the orbitofrontal cortex (OFC) updates the internal

representation of the current context[38,39]. More broadly, a recent meta-analytic approach[36] highlighted the role of the inferior frontal gyrus (IFG) and insula in signaling both prediction errors and predictions. A few model-based fMRI studies investigated hierarchically structured predictions and prediction errors[31–33]. In two associative learning studies[31,32], lower-level prediction errors involved the dopaminergic midbrain, and visual, frontal, parietal, and cingulate regions, while higher-level prediction errors correlated with activity in the cholinergic basal forebrain. In another associative learning task[33], higher-level predictions involved regions such as the OFC or hippocampus, whereas lower-level predictions only triggered activity in the visual cortex. The advantage of these model-based fMRI studies is that they allow to identify predictions or prediction errors at different levels of the hierarchy, on a trial-by-trial basis and with individualized trajectories of the model parameters.

The aim of the present study was to characterize the brain regions involved in signaling hierarchical predictions and prediction errors in NT and autistic adults, and to assess whether there were between-group differences. For this purpose, we used model-based fMRI and the same experimental paradigm as in the study by Weilnhammer et al.[33]. NT and autistic adults performed a crossmodal associative learning task, where a high- or a low-pitch tone was predictive of a clockwise or counterclockwise rotation of two dots, respectively (Fig. 1). In a similar behavioral task, but with a low contingency (i.e., 62.5% of expected outcomes), autistic adults did not update their predictions when the association reversed and were less biased by priors[24]. Here, we used a higher probabilistic association (i.e., 75% as in[33]) to increase the chances that all participants would learn predictions, so that we could identify their neural correlates and pinpoint genuine group differences in neural activation.

The paradigm included three forms of uncertainty. First, sensory uncertainty was introduced with a subset of ambiguous trials where the two dots did not rotate, but simply jumped from their vertical to their horizontal position. In these trials, participants tend to report a rotation direction that is consistent with the main contingency, suggesting that they are biased by their priors[24, 33]. Second, expected uncertainty was introduced by the probabilistic association between the tone and the rotation direction (i.e., only 75% of the trials showed the expected rotation). Third, the task included unexpected uncertainty, as the tone-rotation association could suddenly reverse. We modeled the behavioral data using a three-level Hierarchical Gaussian Filter[40], which offers a way to model perception in uncertain contexts using a hierarchical generative model of the environment. We correlated the trial-wise estimates of the model parameters signaling

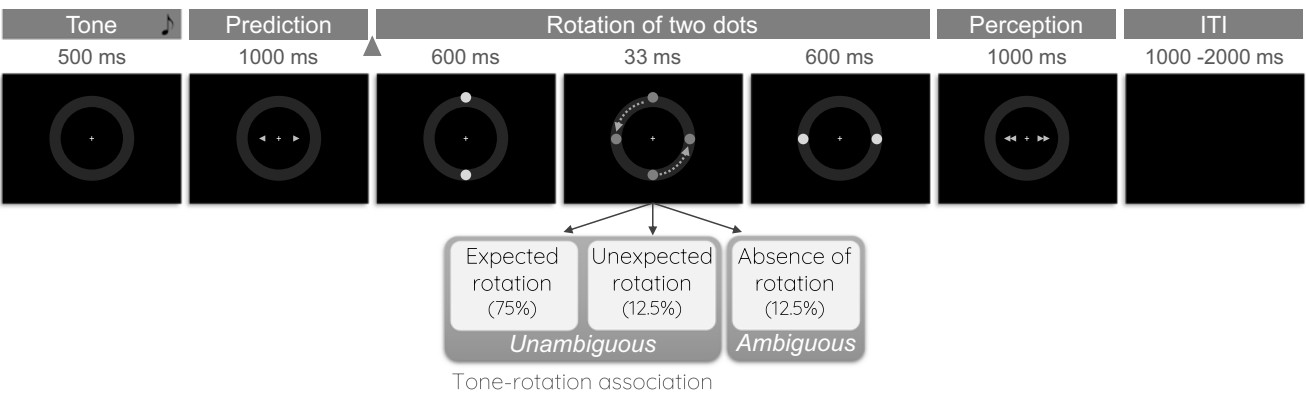

**Fig. 1 | Experimental paradigm.** After hearing a low- or a high-pitch tone, participants had to predict the rotation direction of a pair of dots, and to report their percept. There was a probabilistic association between the tone and the rotation direction (main contingency: 75%), reversing every 16, 24, or 32 trials. In a subset of ambiguous trials, the dots did not rotate but simply appeared in their vertical and then horizontal position. CW: Clockwise, CCW: Counterclockwise, ITI: Inter-trial interval. The triangle indicates a jitter of 100 to 300 ms.

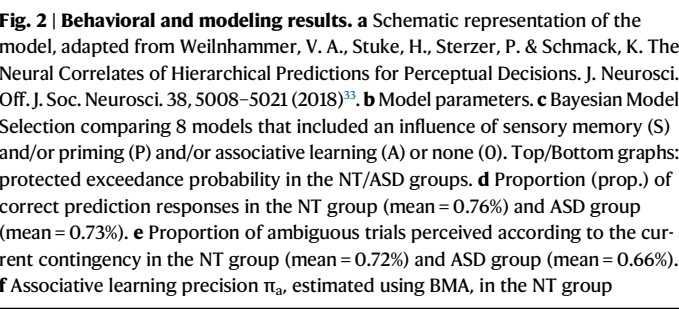

**Fig. 2 | Behavioral and modeling results. a** Schematic representation of the model, adapted from Weilnhammer, V. A., Stuke, H., Sterzer, P. & Schmack, K. The Neural Correlates of Hierarchical Predictions for Perceptual Decisions. J. Neurosci. Off. J. Soc. Neurosci. 38, 5008–5021 (2018)[33]. **b** Model parameters. **c** Bayesian Model Selection comparing 8 models that included an influence of sensory memory (S) and/or priming (P) and/or associative learning (A) or none (0). Top/Bottom graphs: protected exceedance probability in the NT/ASD groups. **d** Proportion (prop.) of correct prediction responses in the NT group (mean = 0.76) and ASD group (mean = 0.73%). **e** Proportion of ambiguous trials perceived according to the current contingency in the NT group (mean = 0.72%) and ASD group (mean = 0.66%). **f** Associative learning precision $\pi_a$, estimated using BMA, in the NT group

(mean = 1.37) and ASD group (mean = 1.32). **g** 2nd level learning rate $\omega_2$, estimated using BMA, in the NT group (mean = −0.72) and ASD group (mean = −0.89). **h** 3rd level learning rate $\omega_3$, estimated using BMA, in the NT group (mean = −6.23) and ASD group (mean = −6.25). **i** Correlation between $\pi_a$ and the percentage of ambiguous trials perceived according to the current contingency. The error band represents the confidence interval. Correlations were assessed using Pearson's correlation test. In the histograms **d**–**h**, error bars indicate standard deviations, and groups were compared using Student's $t$ tests (two-sided, not adjusted for multiple comparisons). Blue: NT participants ($n = 26$). Orange: ASD participants ($n = 26$). ***$p < 0.001$.

predictions and prediction errors with the fMRI time-courses. We also used Dynamic Causal Modeling to investigate how top-down and bottom-up connections were modulated by predictions and prediction errors, respectively. Finally, autistic traits and symptoms were assessed using questionnaires to better characterize our sample and to investigate whether these symptoms could contribute to the group differences.

In NT, we hypothesized that the neural correlates would be in line with the literature[31,33,41]. We predicted a rather large overlap between NT and ASD regarding the set of regions encoding predictions, as several behavioral studies found an intact influence of priors in ASD[21,22,42–45]. Given that in a similar paradigm, autistic individuals updated their predictions less than NT[20], differences might be found in regions involved in learning and adjusting predictions, such as the OFC, IFG, retrosplenial cortex, or (para)hippocampus. In general, group differences are more likely to be found at the higher levels of the hierarchy than at lower levels, as suggested by behavioral results interpreted as an overestimation of the volatility in ASD[26] and by fMRI results showing a similar effect of visual illusions on the primary visual cortex in NT and ASD[46]. The weak prior hypothesis[11] would predict a decreased activation of regions encoding priors, and a less proactive brain in ASD. Specifically, proactive processes at the neural level were evidenced in NT[33] with this paradigm, as activity in the retinotopic areas of the dot trajectories was elicited when hearing a tone. Regarding the neural correlates of prediction errors, the HIPPEA hypothesis would predict a stronger reaction to precision-weighted prediction errors in ASD in regions such as the ACC, caudate nucleus, putamen, basal forebrain, or anterior insula.

## Results

### Behavioral results

Both groups got percentages of correct prediction responses above chance level: 76% (±9) in NT ($t(25) = 14.6$, $p < 0.0001$, $d = 2.86$) and 73% (±11) in ASD ($t(25) = 10.7$, $p < 0.0001$, $d = 2.10$) (Fig. 2d). These percentages did not significantly differ between groups ($t(48) = 0.88$, $p = 0.38$, $d = 0.24$).

Perception responses were analyzed in unambiguous and ambiguous trials, separately. In unambiguous trials, both groups got 99% (±1) of correct perception responses. In ambiguous trials, the percentage of trials perceived as following the current contingency was 72% (±12) in the NT group and 66% (±13) in the ASD group (no significant group difference despite a trend, $t(50) = 1.7$, $p = 0.095$, $d = 0.47$) (Fig. 2e). These percentages were above chance level in the NT group ($t(25) = 9.7$, $p < 0.0001$, $d = 1.90$) and ASD group ($t(25) = 6.6$, $p < 0.0001$, $d = 1.29$), suggesting that they were both biased by their expectations.

Finally, in the confidence rating task, a repeated-measure ANOVA on confidence rating scores showed a main effect of ambiguity ($F(1,50) = 40.0$, $p < 0.0001$), with ambiguous trials being rated as more uncertain than unambiguous trials (60% ± 30 certain in ambiguous trials vs. 90% ± 14 certain in unambiguous trials, $t(51) = 6.4$, $p < 0.0001$, $d = 0.88$), but revealed no group effect nor interaction.

### Behavioral modeling

After inverting the eight models and performing BMS, group-level inference showed that the *Associative learning* model (A) best explained the data in both groups (Fig. 2c). Indeed, the protected

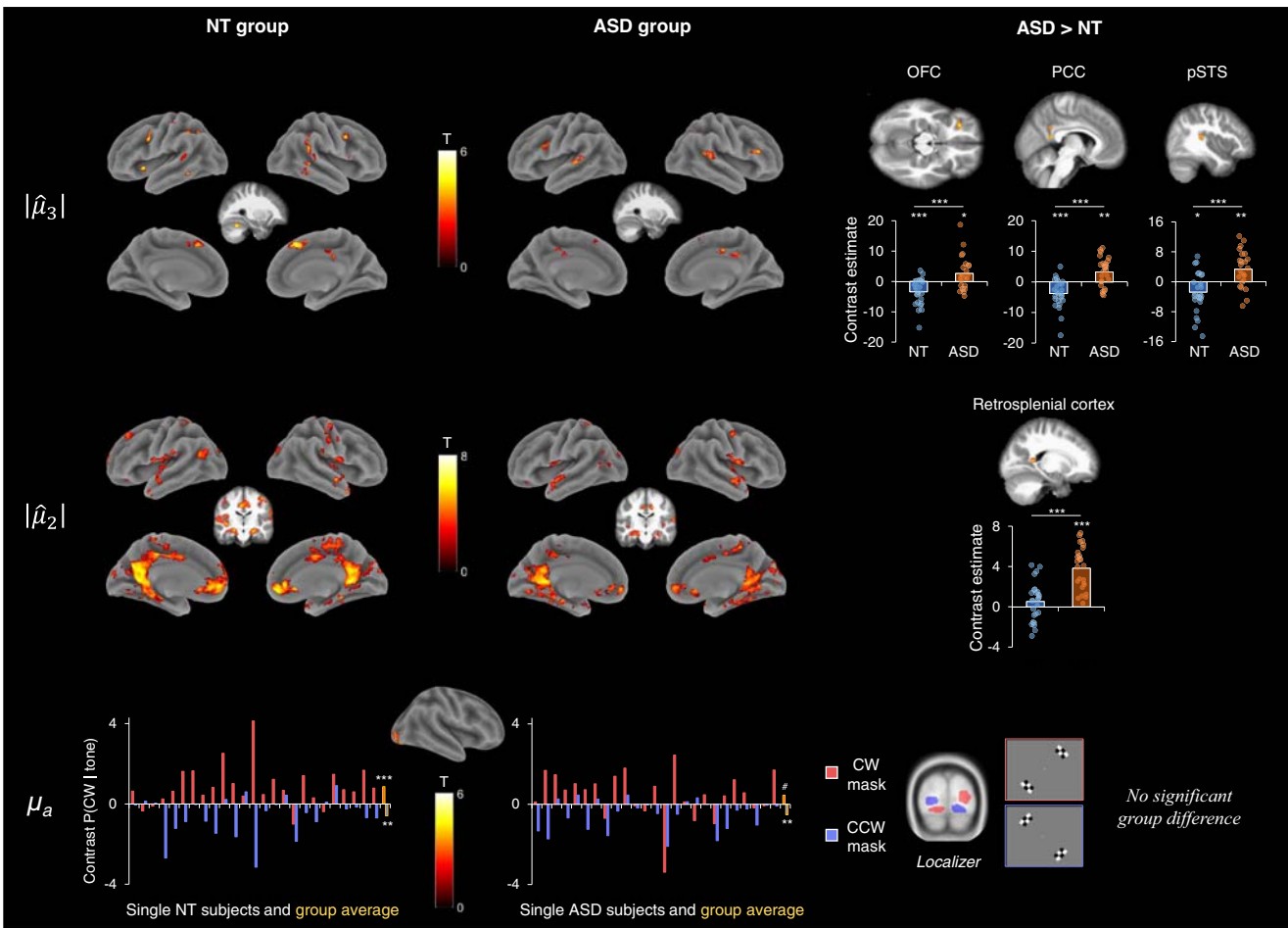

**Fig. 3 | Neural correlates of the prior mean.** Top row: positive effect of the 3rd level prior mean $|\hat{\mu}_3|$ in the NT and ASD groups, and group differences in the left orbitofrontal cortex (OFC), bilateral posterior cingulate cortex (PCC, note that the two PCC clusters were merged into one region for plotting the contrast estimate) and right posterior superior temporal sulcus (pSTS). OFC histogram: one-sample $t$ test in NT: $p < 0.001$, in ASD: $p = .023$; two-sample $t$ test between groups: $p < 0.001$. PCC histogram: one-sample $t$ test in NT: $p < 0.001$, in ASD: $p = 0.002$; two-sample $t$ test: $p < 0.001$. pSTS histogram: one-sample $t$ test in NT: $p = 0.014$, in ASD: $p = 0.002$; two-sample $t$ test: $p < 0.001$. NT group: $n = 26$, ASD group: $n = 25$. Middle row: Positive effect of the 2nd level prior mean $|\hat{\mu}_2|$ in the NT and ASD groups, and group differences in the right retrosplenial complex (overlapping the precuneus and retrosplenial cortex). Retrosplenial histogram: one-sample $t$ test in NT:

$p = 0.182$, in ASD: $p < 0.001$; two-sample $t$ test: $p < 0.001$. NT group: $n = 26$, ASD group: $n = 25$. Bottom row: positive effect of the low-level prior mean $\mu_a$ across groups (central figure) and participants (plots) in the individual masks defined by the localizer. Histograms show the contrast estimates of $\mu_a$ (P(CW|tone)) within individually-defined CW (red) and CCW (purple) masks. The yellow bars on the right of the graphs indicate the group means. NT histogram: one-sample $t$ tests: $p < 0.001$ (CW mask) and $p = 0.006$ (CCW mask). ASD histogram: one-sample $t$ tests: $p = 0.0843$ (CW mask) and $p = 0.002$ (CCW mask). NT group: $n = 25$, ASD group: $n = 25$. In the fMRI analyses, significance level was set at $p < 0.001$ at voxel level and $p < 0.05$ at cluster level. In the histograms, the significance level of the one- or two-sample t tests (two-sided) are indicated as follows: $^{\#}p = 0.08$, $^*p < 0.05$, $^{**}p < 0.01$, $^{***}p < 0.001$.

exceedance probability was 1.00 in the NT group and 0.65 in the ASD group. In the ASD group, the second model that best explained the data was the model *Associative learning & Sensory memory* (AS) (protected exceedance probability of 0.35).

Following BMA, the precision of associative learning $\pi_a$ was 1.37 ($\pm0.28$) in the NT group and 1.32 ($\pm0.28$) in the ASD group (no significant group difference: $t(50) = 0.6$, $p = 0.52$, $d = 0.18$) (Fig. 2f). The second-level learning rates $\omega_2$ were $-0.72$ ($\pm0.86$) in NT and $-0.89$ ($\pm1.28$) in ASD (no significant group difference: $t(44) = 0.5$, $p = 0.59$, $d = 0.15$) (Fig. 2g). The third-level learning rates $\omega_3$ were $-6.23$ ($\pm0.09$) in NT and $-6.25$ ($\pm0.11$) in ASD (no significant group difference: $t(48) = 0.6$, $p = 0.56$, $d = 0.16$) (Fig. 2h). The estimates of the other posterior parameters are given in the Supplementary Table S1.

As expected for a successful inversion of our model, the strength of associative learning $\pi_a$ was positively correlated with the percentage of correct prediction responses ($r = 0.50$, $p < 0.001$) and of ambiguous trials perceived according to the current contingency ($r = 0.66$, $p < 0.001$, Fig. 2i).

We investigated whether the strength of associative learning was correlated with autistic traits (AQ), intolerance of uncertainty (IU), and atypical sensory sensitivity (GSQ). After correcting for multiple comparisons using FDR correction, none of these correlations were significant on the entire sample or within the ASD group, but two correlations remained significant in NT. Within the NT group, the strength of associative learning $\pi_a$ was negatively correlated with the number of autistic traits (AQ, $r = -0.54$, $p_{FDR\text{-}corr} < 0.01$) and with the atypical sensory sensitivity (GSQ, $r = -0.54$, $p_{FDR\text{-}corr} < 0.01$).

## fMRI preamble

After having identified that the associative learning model best explained the behavioral data in both groups, we aimed at characterizing the neural correlates of prior mean and precision as well as prediction errors at several levels of the hierarchy in NT and ASD participants. In all the analyses reported below, the statistical threshold was set at $p < 0.001$ at voxel level and $p < .05$ at cluster level, unless

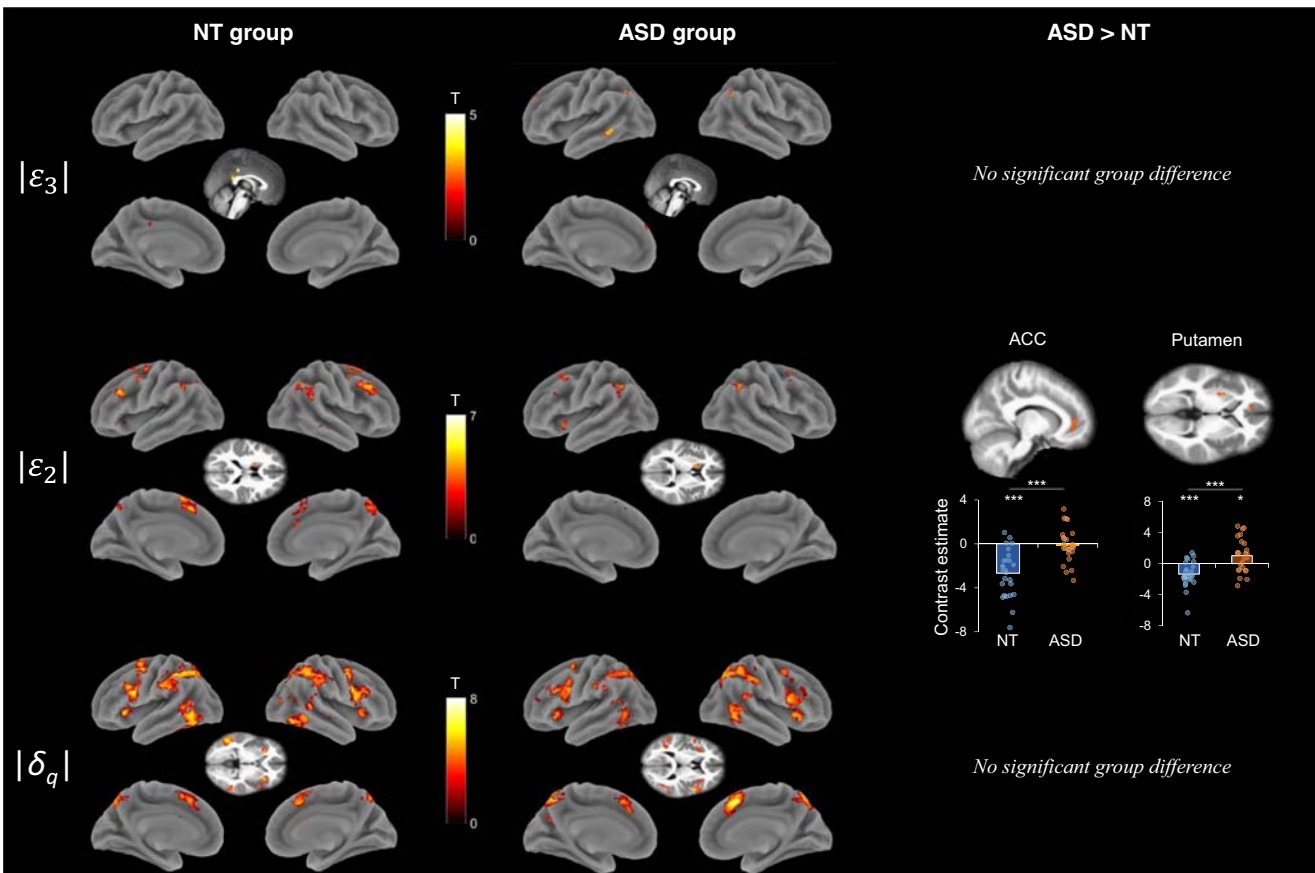

**Fig. 4 | Neural correlates of the prediction errors.** Top row: positive effect of the 3rd level precision-weighted prediction error $|\varepsilon_3|$ in NT and ASD. Middle row: positive effect of the 2nd level precision-weighted prediction error $|\varepsilon_2|$ in NT and ASD, and group differences in the left anterior cingulate cortex (ACC) and left putamen. ACC histogram: one-sample $t$ test in NT: $p < 0.001$, in ASD: $p = 0.709$; two-sample $t$ test: $p < 0.001$. Putamen histogram: one-sample $t$ test in NT: $p < 0.001$, in ASD: $p = 0.029$; two-sample $t$ test: $p < 0.001$. Bottom row: Positive effect of the absolute perceptual prediction error $|\delta_q|$ in the NT and ASD groups. Significance level was set at $p < 0.001$ at voxel level and $p < 0.05$ at cluster level. In the histograms, the significance level of one- and two-sample $t$ tests (two-sided) are indicated as follows: * $p < 0.05$, *** $p < 0.001$. NT group: $n = 26$, ASD group: $n = 25$.

it is specified "$p_{FWE\text{-}corr} < 0.05$", in which case FWE correction was applied at cluster level.

As expected, the contrast *tone* vs. *baseline* showed significant activation in the auditory cortex (superior temporal cortex, Heschl gyrus), precentral cortex, superior frontal, and superior parietal gyri in both groups ($p_{FWE\text{-}corr} < 0.05$). The contrast *rotation* vs. *baseline* involved the bilateral occipital cortex, precentral gyri, cerebellum and putamen in both groups, as well as the superior frontal gyrus in NT and superior parietal cortex in ASD ($p_{FWE\text{-}corr} < 0.05$). There were no significant group differences for these two contrasts. The main focus of the analyses was on the correlates of the estimated model parameters that were added as parametric modulators of the *tone* and *rotation* regressors, as detailed below.

### Neural correlates of predictions

The correlates of predictions (high-level: $|\widehat{\mu_3}|$, mid-level: $|\widehat{\mu_2}|$ and low-level: $\mu_a$) are illustrated in Fig. 3 and detailed in Supplementary Table S2.

In NT, high-level predictions $|\widehat{\mu_3}|$ (i.e., strength of the belief about the volatility) involved the cerebellum, supplementary motor area (SMA), precentral gyrus, middle frontal gyrus, insula, posterior superior (pSTS) and middle temporal gyri and inferior parietal lobe ($p_{FWE\text{-}corr} < 0.05$). In ASD, high-level predictions involved the SMA, pSTS, IFG, and middle cingulate cortex ($p_{FWE\text{-}corr} < 0.05$). In comparison with NT, the ASD group showed a stronger correlation between

$|\widehat{\mu_3}|$ and the activity in the left OFC, bilateral posterior cingulate cortex and right pSTS.

In both groups, mid-level predictions $|\widehat{\mu_2}|$ (i.e., strength of the belief about the probabilistic association) involved a broad cluster covering several medial and central regions (including the cingulate cortex, SMA, medial frontal, OFC, precuneus, lingual gyrus, cerebellum, striatum, hippocampus, parahippocampus), as well as the pre- and postcentral gyri, auditory cortex, pSTS and temporal pole ($p_{FWE\text{-}corr} < 0.05$). The correlation between $|\widehat{\mu_2}|$ and activity in the right retrosplenial complex was stronger in ASD than NT.

Low-level predictions $\mu_a$ were associated with activity in the occipital cortex and left postcentral gyrus (whole-brain analysis) and were further explored through a ROI-analysis. Using the individual masks mapping the CW and CCW rotations, we extracted the contrast estimate $\mu_a$ corresponding to the probability of having a CW rotation given the tone. As displayed in Fig. 3, for most of the participants, hearing a tone that was predictive of a CW or CCW rotation yielded activity in the CW or CCW mask, respectively. In the CW mask, it was the case for 84% of NT and 68% of ASD participants, and in the CCW mask, for 72% of the participants in each group (no significant proportion differences). Across NT participants, the mean activation level was significantly different from zero for both the CW ($t(24) = 4.2$, $p < 0.001$, $d = 0.84$) and CCW ($t(24) = 3.0$, $p < 0.01$, $d = 0.61$) masks. Across ASD participants, the mean activation level was significantly different from zero for the CCW mask ($t(24) = 3.4$, $p < 0.01$, $d = 0.68$) but not for the CW mask ($t(24) = 1.8$, $p = 0.08$, $d = 0.36$).

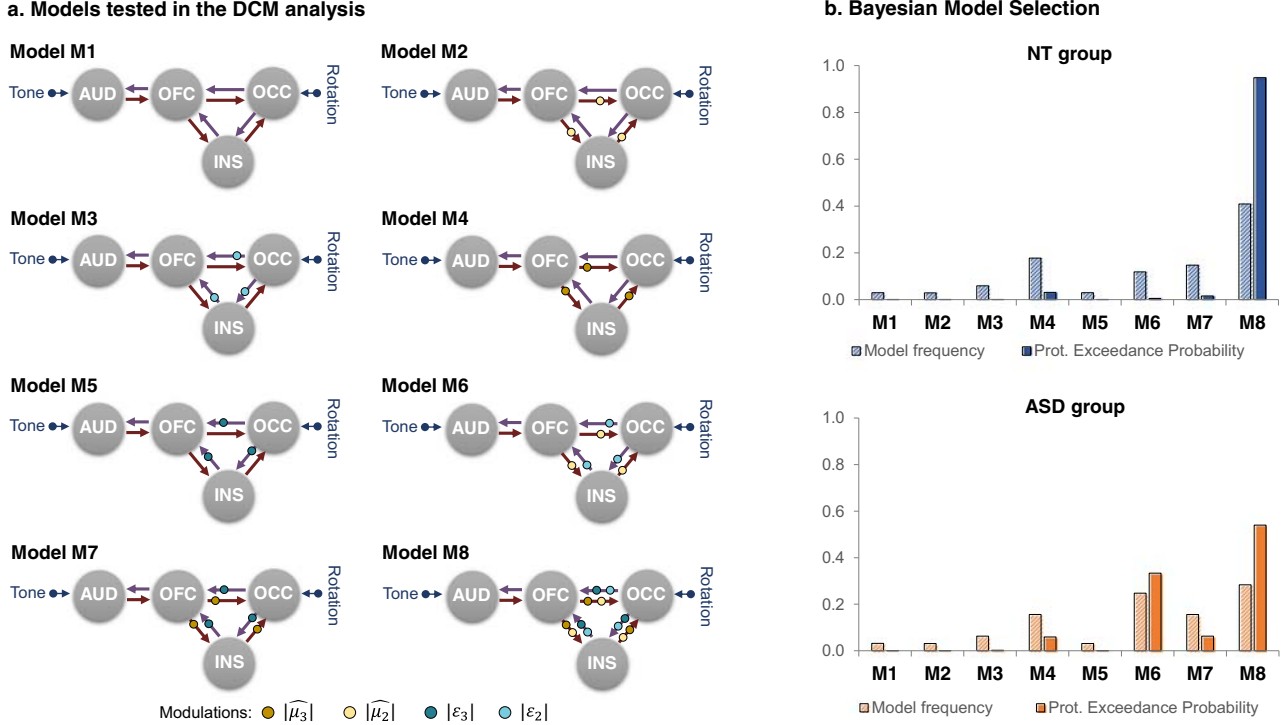

**Fig. 5 | Dynamic causal modeling. a** Description of the eight models used in the Bayesian Model Selection (BMS) assessing the effect of mid- and high-level priors and precision-weighted prediction errors on top-down and bottom-up connections. *AUD* left auditory cortex, *INS* left insula, *OFC* left orbitofrontal cortex, *OCC* occipital cortex. To simplify the figure, the OCC_L and OCC_R were represented here as a single ROI, but the OCC_L and OCC_R were modeled separately. **b** Result of the BMS, displaying the model frequency and protected (prot.) exceedance probability.

## Neural correlates of the prediction precision

The regions whose activity correlated with the precision of mid- and high-level predictions are listed in Supplementary Table S3 and shown in Supplementary Figure S1. The precision of high-level predictions $\widehat{\pi_3}$ correlated with activity in the bilateral medial occipital cortex, middle and superior temporal cortex and temporo-parietal junction (TPJ) in both groups, with the bilateral middle cingulate cortex and right IFG in NT, and with the left postcentral gyrus, right central sulcus and bilateral cerebellum in ASD ($p_{FWE\text{-}corr} < 0.05$). Activity in the left superior frontal gyrus was more strongly correlated with $\widehat{\pi_3}$ in NT than ASD.

The precision of mid-level predictions $\widehat{\pi_2}$ did not yield any significant activity in any group. The only region that was close to significance level ($p < 0.001$ at voxel level, $p < 0.08$ at cluster level) was the right parahippocampal cortex in the ASD group. In comparison with NT, activity in the right superior parietal gyrus was more strongly correlated with $\widehat{\pi_2}$ in ASD than NT.

## Neural correlates of prediction errors

Finally, we were interested in characterizing the neural correlates of absolute precision-weighted prediction errors $|\varepsilon_3|$ and $|\varepsilon_2|$, and perceptual prediction errors $|\delta_q|$. The results are detailed in Supplementary Table S4 and shown in Fig. 4.

High-level precision-weighted prediction errors were associated with activity in the bilateral middle/posterior cingulate cortex in NT, and in the bilateral middle temporal gyrus, bilateral angular gyrus, and left superior frontal gyrus in ASD. There were no significant group differences.

Mid-level precision-weighted prediction errors correlated with activity in the bilateral SMA, middle and superior frontal gyrus, and inferior parietal lobe in both groups ($p_{FWE\text{-}corr} < 0.05$). In addition, they were related to the bilateral superior medial frontal gyrus, middle cingulate cortex, and precuneus in NT, and to the left insula in ASD

($p_{FWE\text{-}corr} < 0.05$). The correlation between $|\varepsilon_2|$ and the activity level in the left ACC and putamen was higher in ASD participants than NT. Note that the group difference in the ACC actually denotes an absence of correlation in the ASD group.

Perceptual prediction errors gave a similar pattern in both groups, with an involvement of the bilateral precentral gyrus, IFG, SMA, parietal lobe, insula, posterior inferior/middle temporal gyrus ($p_{FWE\text{-}corr} < 0.05$). There were no significant group differences.

## Correlations between fMRI results and questionnaire scores

We assessed correlations between the contrast estimates in regions where there was a group difference and the scores of the questionnaires assessing autistic traits or symptoms. Given the presence of a group difference in these two domains, a correlation was expected, but this was performed in order to evaluate whether certain symptoms (e.g., sensory sensitivity) were especially related to one of the group differences. Medium to large positive correlations were found with most of the questionnaires, especially with the IU and AQ, and the results are presented in Supplementary Table S5.

## DCM analyses

We assessed whether top-down and bottom-up connections were modulated by high-level and/or mid-level predictions and/or precision-weighted prediction errors, respectively (Fig. 5a). The BMS (Fig. 5b) revealed that M8 was the best model in the NT group (protected exceedance probability: 0.95, model frequency: 0.41). In the ASD group, the results were less clear, with M8 being the best model (protected exceedance probability: 0.54, model frequency: 0.28), followed by M6 (protected exceedance probability: 0.33, model frequency: 0.25).

A BMA on the posterior parameters showed that all the intrinsic connections were significantly different from zero ($p$ values < 0.001 in

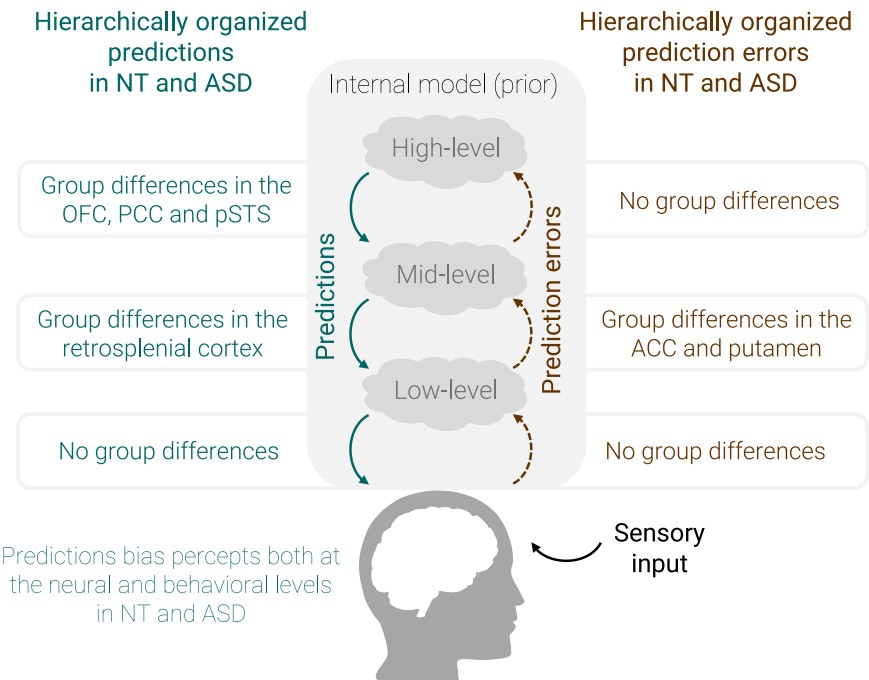

**Fig. 6 | Summary figure.** Using an associative learning task and model-based fMRI, we characterized the neural correlates of hierarchically structured predictions and prediction errors in neurotypical (NT) and autistic (ASD) adults at the low, mid, and high levels of the hierarchy. Both groups managed to make accurate predictions and were biased by their expectations both at the behavioral and neural levels. The neural networks identified in the two groups were globally similar, but group differences were found in a set of brain regions. High-level predictions were following activity levels in the left orbitofrontal cortex (OFC), bilateral posterior cingulate cortex (PCC), and right posterior superior temporal sulcus (pSTS) more closely in ASD than NT. The right retrosplenial cortex was more strongly correlated with mid-level predictions in ASD than NT. Mid-level prediction errors were correlated with activity in the anterior cingulate cortex (ACC) in NT but not in ASD, and were more strongly correlated with activity in the left putamen in ASD.

both groups), except for the connection from the INS to the OFC (NT: $p = 0.10$, ASD: $p = 0.52$). In NT, there were modulatory effects by $|\varepsilon_3|$ on the connections from the $OCC_L$ and $OCC_R$ to the INS ($p$ values $< 0.005$), and by $|\varepsilon_2|$ on the connections from the $OCC_R$ to the INS ($p$ values $< 0.05$). In NT, there was a modulatory effect by $|\widehat{\mu_2}|$ on connections from the OFC to the $OCC_L$ ($p < 0.01$), and non-significant trends for modulations by $|\widehat{\mu_3}|$ on connections from the OFC to the $OCC_L$ ($p = 0.067$) and by $|\widehat{\mu_2}|$ on the connections from the INS to the $OCC_L$ ($p = 0.099$). In ASD, there were modulatory effects by $|\varepsilon_3|$ on the connections from the $OCC_L$ to the INS ($p < 0.05$) and to the OFC ($p < 0.05$). In ASD, there was a modulatory effect by $|\widehat{\mu_3}|$ on connections from the OFC to the INS ($p < 0.005$), and a non-significant trend toward a modulation by $|\widehat{\mu_2}|$ on this connection ($p = 0.062$). Between-group comparisons only showed a non-significant trend toward a group difference on the modulation by $|\varepsilon_2|$ on the connection from the $OCC_L$ to the INS ($t(48) = 1.7$, $p = 0.89$, NT: 0.02 Hz ±0.18, ASD: −0.16 Hz ± 0.48).

## Discussion

In this study, we used model-based fMRI to investigate the neural correlates of predictive mechanisms in ASD. We were particularly interested in characterizing the set of brain regions involved in signaling predictions and prediction errors in ASD, and in potentially revealing group differences to shed light on Bayesian theories of ASD. At the behavioral level, the two groups managed to make accurate predictions and were biased by their expectations. At the neural level, we identified the correlates of hierarchical predictive processes within group and found a rather large overlap of brain regions between groups, but also some group differences at the higher levels of the hierarchy. The main findings of this study are summarized in Fig. 6.

The predictive skills of ASD participants appeared to be typical in this associative learning task. Indeed, both groups managed to learn predictions and were biased by their expectations, which indicates that autistic individuals do not have uniformly weak priors. In this volatile environment, autistic adults managed to flexibly use prior knowledge to make perceptual inference. In the same task with a more uncertain context (i.e., lower probabilistic association)[24], autistic adults were also able to learn predictions, but failed to update them after a change in contingency. As compared to NT, autistic adults had a decreased precision of associative learning ($\pi_a$) and were less biased by priors[24]. This is in contrast with the current study where no such group differences were found, even though more autistic traits in NT were also associated with lower $\pi_a$. It suggests that autistic adults are able to handle some degrees of expected and unexpected uncertainty, but might be impaired above a certain level of uncertainty. This is in line with a recent review[20] concluding that associative learning may be particularly impaired when predictive features have low consistency. Similarly, a probabilistic reinforcement learning task[47] showed that autistic adults managed to learn stimulus pairs that were frequently reinforced (80%), but were impaired when the reinforcement was less frequent (70%). In addition, we found no group differences in learning rates, consistently with the behavioral experiment cited above[24]. In some other associative learning tasks, the learning rates did not differ either in autistic vs. typically developing children[25], or in adults with more or less autistic traits[48].

The neural correlates of predictions were identified in each group, and to the best of our knowledge, these regions had not been characterized in ASD before. First, we observed a hierarchical organization of regions encoding predictions in each group. While low-level predictions only involved the occipital cortex, higher-level predictions elicited activity in a broader set of supramodal regions. The regions identified along this hierarchy resemble the principal gradient of macroscale organization in the human connectome[49], as higher-level predictions involve regions belonging to the principal gradient, i.e., transmodal regions including the default mode network and

frontoparietal regions, while low-level predictions resemble the map of the second functional gradient. Regions of the default mode network are at the upper end of the topographical hierarchy and integrate information from several sensory modalities to get abstract representations[49]. This functional gradient from primary sensory regions to transmodal regions is therefore in line with the hierarchical neural correlates of predictions that we identified. Interestingly, a recent study investigated the microstructural profiles of these regions[50]. They identified gradual cytoarchitectural variations that globally followed the functional gradient, but the transmodal cortices had a less hierarchical organization[50]. The rich and less-constrained interconnectivity of these transmodal regions would be central to underlie flexible cognitive functions[50], which may allow representing higher-level abstract representations, for instance about the stability of predictions over time. Interestingly, at the lower-level of the hierarchy, we found that hearing a tone elicited activity in the retinotopic areas representing the CW or CCW rotations that was modulated specifically by the rotation direction predicted by the tone. This effect is consistent with findings in NT[33], and reveals proactive mechanisms in both NT and autistic individuals. Together with the prior bias observed at the behavioral level, this fMRI result indicates a similar influence of priors on sensory processing at low hierarchical levels in autistic individuals as in NT. In particular, using fMRI allowed to point out that sensory processing itself is biased by expectations in ASD, resolving an ambiguity, which often plagues behavioral studies, on whether priors bias perceptual or decisional processes. Hence, in the context of this experiment, the findings do not support the weak prior hypothesis of ASD[11]. This result for low-level predictions is in line with a recent study showing that the illusory triangle induced in the primary visual cortex by the Kanizsa figure did not differ between NT and ASD[46], suggesting an intact perceptual bias in ASD. Future studies including more ambiguous trials could perform decoding analyses (e.g., such as in ref. 51) to assess if the perceptual bias induced in ambiguous trials generates activity in motion area MT.

The pattern of brain regions associated with mid-level predictions is highly consistent with the literature[34,36,52–54], in particular with results in NT using the same paradigm[33] and showing an involvement of the OFC, hippocampus, insula, precuneus, and medial frontal gyrus. This pattern is also consistent with regions that are processing contextual associations, such as the retrosplenial complex and parahippocampal cortex, and that contribute to eliciting prediction-related representations in the OFC[1]. In particular, the activity level in a cluster of the right retrosplenial complex followed the strength of mid-level predictions more closely in ASD than NT. This region plays a fundamental role in learning contextual associations between sensory stimuli[1,55–57]. One potential explanation for this group difference could be that the tone-rotation association was learned more strongly or more rigidly, in ASD. This stronger correlation might also be interpreted as ASD participants needing a greater involvement of this region to reach a similar performance. Future studies could use a causal approach to further corroborate these interpretations.

High-level predictions about the stability of the association over time evoked activity in regions such as the middle/posterior cingulate gyrus, IFG, pSTS or SMA in both groups. Yet, the pattern of regions seemed a bit more distinct between groups than for mid-level predictions. In particular, there were stronger correlations between high-level predictions and activity in the left OFC, bilateral posterior cingulate cortex and right pSTS in ASD, relative to NT. A plausible hypothesis could be that these regions might encode the volatility more precisely in ASD, which may be related to the idea of an overestimation of the environmental volatility in ASD[26]. It may also contribute to a higher intolerance of uncertainty and more autistic traits, as suggested by the correlations with the questionnaires. The stronger correlations with high-level predictions in these regions might entail different predictive processes in ASD. For instance, the posterior

cingulate cortex signals the expected environmental information and contributes to increase routines[58]. Particularly, the OFC plays a key role in encoding prior uncertainty, maintaining contextual priors and updating perceptual beliefs[1,34,59,60]. In ASD and NT, the model best explaining the data was the one where both mid- and high-level predictions modulating top-down connections from the OFC to the visual cortex and insula. We can hypothesize that the stronger activation of the OFC in ASD may be due to more top-down prior knowledge sent from this region in ASD, which might suggest a difference in the specialization of the OFC to encode priors. Nevertheless, note that the extent to which predictions modulated top-down connections from the OFC did not differ significantly between groups.

Regarding prediction errors, the neural correlates of low-level perceptual prediction errors encompassed regions such as the IFG, insula, precentral gyrus, inferior parietal and the middle temporal gyrus in both groups, consistently with findings in NT[33]. As reported in a recent meta-analytic study[36], the IFG and insula were found to be involved in signaling both predictions and prediction errors. The absence of group differences suggests that low-level perceptual prediction errors may be processed similarly in NT and ASD. Mid-level precision-weighted prediction errors involved, among others, middle and superior frontal regions, the caudate nucleus, the insula and inferior parietal regions in both groups, which is again in line with results in NT[31,33]. In ASD compared to NT, there was a stronger correlation between mid-level prediction errors and activity levels in the putamen, a region encoding prediction errors[41,61]. A plausible interpretation could be a higher weight of mid-level prediction errors in ASD, in line with the HIPPEA theory[13] suggesting that highly precise prediction errors would cause overwhelming sensations of surprise, leading them to avoid unpredictable environments. Furthermore, we observed a negative correlation between neural activity in the ACC and mid-level precision-weighted prediction errors in NT, but not in ASD. Interestingly, in a social context, the gyral surface of the ACC showed a deactivation in response to social prediction errors in NT, but not in ASD[28]. Moreover, they found that the activation level of the ACC was positively correlated with the social symptoms of ASD assessed with a sub-scale of the ADOS[28], while we also found positive correlations between the ACC activity level and autistic traits or social difficulties across groups. The encoding of prediction errors in the ACC might therefore be altered across multiple domains in ASD, and could play a key role in the atypical predictive learning mechanisms observed in ASD. Finally, even though no significant group differences were found, high-level precision-weighted prediction errors were associated with distinct sets of regions in each group. The DCM analysis showed that both high- and mid-level precision-weighted prediction errors modulated bottom-up connections similarly in each group.

The current study has several limitations. First, the DCM analysis only focused on a reduced set of regions and might have shown a different pattern of results if we would have selected other candidate regions such as the IFG, ACC, caudate nucleus, or putamen. Furthermore, in the fMRI analyses, we decided to focus on positive effects of the model quantities on brain activity, which seems easier to interpret, but negative effects may also provide additional information. Using a design with shorter TRs could have revealed variability across different regions of the cortex in the temporal shape of the HRF[62], with faster responses in sensory regions than in associative areas[63]. We chose to rely on a dual-report paradigm with both prediction and perception responses to get an explicit measure of prediction learning (where tone could not be simply ignored) and to get an implicit measure of perceptual bias, but using such paradigm might bias responses as participants often tend to ensure self-consistency[64]. Finally, some of our autistic participants had comorbidities or were taking

**Table 1 | Demographic characteristics and questionnaire scores of the participants**

|  | NT group | ASD group | *p* |
|---|---|---|---|
| Number of participants | 26 | 26 | - |
| Male/female number | 13/13 | 13/13 | ns |
| Age (years) | 30.9 (±8.3) | 32.2 (±9.5) | ns |
| Left/right-handed | 2/24 | 5/21 | ns |
| Total IQ score | 113.9 (±12.3) | 112. 1 (±16.5) | ns |
| AQ score | 13.0 (±6.2) | 32.1 (±8.7) | *** |
| IU score | 27.9 (±8.4) | 42.7 (±6.6) | *** |
| GSQ score | 34.5 (±16.7) | 53.3 (±21.2) | *** |

The table presents the group means (±standard deviations). *IQ* Intelligence Quotient (Wechsler Adult Intelligence Scale IV), *AQ* Autism spectrum Quotient, *IU* Intolerance of Uncertainty, *GSQ* Glasgow Sensory Questionnaire. Note that IQ data from four NT participants are missing as they were not native Dutch speakers. One of the ASD participants was included in the behavioral analyses but not in the fMRI analyses. The two groups were compared using two-sided Student *t* tests (i.e., age, total IQ, AQ, IU, and GSQ scores) and chi-square tests (i.e., proportions of male/female, left/right-handed). *ns* non-significant, ****p* < 0.001.

medications, which may have influenced their behavior and neural responses. Yet, we believe that including such participants is more representative of the ASD population, as having comorbidities or taking some medication is highly frequent in ASD.

In conclusion, we found that autistic adults can learn priors and tend to encode priors hierarchically, like NT. Yet, they showed alterations in the neural processing of mid- and high-level predictions that may be interpreted as a stronger encoding of these predictions. In addition, as autistic individuals had their percepts biased by their expectations both at the behavioral and neural levels, it does not support the weak prior hypothesis of ASD[11]. Correlates of mid-level precision-weighted prediction errors may be in line with stronger prediction errors in ASD, consistent with the HIPPEA hypothesis[13]. Overall, the predictive mechanisms appear to be relatively intact in ASD, but subtle differences at higher levels of the hierarchy might contribute to the difficulties in predictive learning observed in certain contexts. These results suggest that the Bayesian account of ASD should be refined to better describe predictive mechanisms in ASD. Finally, planning comparative neuroimaging studies with other clinical populations hypothesized to have atypical perceptual inference processes (e.g., schizophrenia or dyslexia) would help to better identify the specificities of each condition regarding predictive skills.

## Methods
### Participants
Twenty-six neurotypical (NT) adults and 26 autistic adults participated in the MRI experiment. Their demographic characteristics are shown in Table 1. The two groups were matched for age, sex ratio, handedness ratio, and total intelligence quotient. One autistic participant completed the behavioral experiment but fMRI scans were not acquired as she was too sensitive to the noise of the scanner when we started the fMRI acquisition. Hence, the sample size is 26 in each group for the behavioral/model analyses, but 25 in the ASD group, and 26 in the NT group for the fMRI analyses. Note that for the localizer analyses, the sample size is 25 ASD and 25 NT participants, as there was a technical failure during the acquisition of the localizer of one NT participant.

Inclusion criteria were being between 18 and 50 years old, reporting normal or corrected-to-normal hearing and vision. Exclusion criteria were having contra-indication for MRI, having a total intelligence quotient below 70 at the Wechsler Adult Intelligence Scale IV[65], or scoring above 32 at the Autism Spectrum Quotient (AQ)[66] for NT participants. Autistic participants received their diagnoses from a multidisciplinary Expertize Center for Autism (University Hospitals of KU Leuven) in a standardized way according to

the criteria of the DSM-5[15], and had idiopathic ASD. Autistic adults were recruited via this expertize center and via the LAuRes (Leuven Autism Research) consortium website. NT participants were recruited via the University of Leuven or acquaintances. None of the NT participants reported having a comorbidity or being under medication. Five ASD participants reported having one or several comorbidities (ADHD (4), dyslexia (2), Gilles de la Tourette (1)). These participants who had comorbidities are included in the analyses, but note that the results of the behavioral and fMRI analyses did not change after removing these five participants. Eleven ASD participants reported taking one or several medications (Abilify (2), Asaflow (1), Bufonix (1), Celecoxib (1), Deanxit (1), Depakine (1), Escitalopram (1), Fluoxetine (1), Fluoxone (1), Hydrea (1), L-Thyroxine (3), Medikinet (1), Melatonine (1), Montelucast (1), Notrilen (1), Redomex (1), Ritalin (1), Trazadone (1), Venlafaxine (1), Welbutrin (1)).

The study was approved by the medical Research Ethical Committee UZ / KU Leuven. All participants provided written informed consent, according to the Declaration of Helsinki.

### Overall procedure
Prior to the experiment, participants filled out online questionnaires (in Dutch or English, depending on their native language): AQ[66,67], short version of the Intolerance of Uncertainty scale[68,69] and Glasgow Sensory Questionnaire[70,71]. Participants started with a short training (7 trials), before being installed in the MR scanner. After a T1-weighted anatomical scan, participants performed another short training and five fMRI runs, followed by a localizer run. Magnetic resonance spectroscopy data were also acquired during this session, as part of another study[72].

Data were acquired on a 32 head coil 3 T Philips Achieva system at the University Hospital of Leuven. During the fMRI acquisition, stimuli were projected on a screen behind the scanner, and reflected through a mirror mounted on the head coil. Stimuli were presented using the Psychtoolbox (version 3) in Matlab (version 2019a), and auditory stimuli were presented binaurally via headphones.

### Experimental paradigm
#### Main task
**Trial structure.** Participants performed an associative learning task (Fig. 1, based on[33]) where tones were probabilistically associated with the rotation direction of two dots. Participants were instructed that there was an underlying association between the tone and the rotation direction, and that this association could change. A high (576 Hz) or low (352 Hz) tone was presented for 500 ms and followed by a *prediction* response screen for 1000 ms (jitter of 100 to 300 ms between the tone and the prediction screen). The prediction screen displayed a right and a left arrow, and participants had to click on the right or left button of the MRI response box if they thought that the tone was predictive of a clockwise (CW - right) or counterclockwise (CCW - left) rotation, respectively. The arrow selected by the participant turned red. Then, two dots appeared at their vertical position for 600 ms, made a CW or CCW rotation within 33 ms and remained at their horizontal position for 600 ms. This was followed by a *perception* response screen showing a right and a left double arrow, displayed for 1000 ms. Participants had to report whether they perceived a CW or CCW rotation using the right or left button, respectively (the selected double-arrow turned red). The inter-trial interval lasted for 1000 to 2000 ms (uniform distribution).

There were two types of trials: *unambiguous trials* where the two dots rotated, and *ambiguous* trials where the pair of dots simply appeared in their vertical and then horizontal positions (no rotation). While unambiguous trials allowed learning the contingency, ambiguous trials allowed assessing whether participants were biased by their expectations (i.e., if they would report a rotation consistent with their expectations, despite the absence of rotation).

**Run structure.** Participants completed five runs of 72 trials (total of 360 trials). In each run, there were 12.5% of ambiguous trials (9 trials) and 87.5% of unambiguous trials (63 trials). Among the unambiguous trials, 75% were expected (i.e., main tone-rotation association) and 12.5% were unexpected (i.e., least frequent association). The trial order was pseudo-randomized so that these percentages (75% unambiguous expected, 12.5% unambiguous unexpected, 12.5% ambiguous) remained the same across eight successive trials. In each run, the contingency reversed after 16, 24, or 32 successive trials.

**Localizer task.** The localizer task was designed for retinotopic mapping of the CW and CCW dot trajectories. Checkerboards covering the dot CW or CCW trajectory were displayed for 15 s, and flickered at a frequency of 8 Hz. The checkerboards did not overlap the initial vertical or final horizontal dot position but covered the upper-right and lower-left quadrants in CW trials, and the upper-left and lower-right quadrants in CCW trials. There were 4 CW trials and 4 CCW trials interleaved, separated by 5 s of fixation. To maintain attention, participants had to fixate on a central cross and click on the left button when the cross changed of color (white or red).

**Post-fMRI confidence rating task.** After finishing the fMRI experiment, participants completed a short computer task as in[33] to assess the perceptual quality of the ambiguous trials. The structure of this task was the same as the main task, but trials included a third response screen showing the options "*1. Very sure*", "*2. Quite sure*", "*3. Quite unsure*", "*4. Very unsure*" (displayed for 2600 ms). Participants used the numbers 1 to 4 to indicate how confident they were about their perception response. There were 48 trials, including 50% of ambiguous trials and 50% of unambiguous trials. To calculate a mean confidence rating, the 1 (*Very sure*) to 4 (*Very unsure*) scale was transformed into a 100% to 0% certainty scale.

### MRI acquisition
**Anatomical scan.** A high-resolution T1-weighted anatomical scan was acquired with a MPRAGE sequence (200 contiguous coronal slices, voxel size = $1 \times 1 \times 1$ mm$^3$, TR = 9.7 ms, TE = 4.6 ms, field of view = $256 \times 240 \times 200$ mm$^3$, acquisition matrix = $256 \times 238$, acquisition time = 4 min 35 s).

**Functional MRI.** Whole-brain T2*-weighted echo-planar imaging sequences were collected (voxel size = $2 \times 2 \times 2$ mm$^3$, TR = 2 s, TE = 30 ms, flip angle = 90°, FOV = $224 \times 224 \times 132$ mm$^3$, 60 ascending transverse slices, multi-band factor: 2). In each run of the main task, 198 volumes were acquired (6 min 44 s per run, total of 990 volumes). In the localizer run, 160 volumes were collected (4 min 52 s).

### Behavioral analyses
**Statistical analyses.** All the results are presented as mean (±standard deviation). Demographic data (Table 1) of the NT and ASD groups were compared using Student's $t$ tests and chi-square tests. The percentage of correct predictions was compared to chance level using one-sample $t$ tests with μ = 0.50. Between-group comparisons on accuracy levels were performed using two-sample $t$ tests. Effect sizes are reported as Cohen's $d$: very small ($d = 0.01$), small ($d = 0.20$), medium ($d = 0.50$), large ($d = 0.80$), or very large ($d > 1.20$) effect sizes[73,74]. Correlations were assessed using Pearson's correlation test, and were corrected for multiple comparisons using False Discovery Rate (FDR) correction. Statistical analyses were performed using R (version 4.0.3, http://www.r-project.org/). All Student's t tests were two-tailed. The threshold for statistical significance was set at $p < 0.05$.

**Computational modeling.** The behavioral data were modeled using the Bayesian modeling approach of Weilnhammer and colleagues[33] (see the Supplementary Note S1 for the full mathematical model description). The model consists of a *contingency model* and a *perceptual model* (Fig. 2a–b). The mean and variance of the priors used in this model are shown as Supplementary Table S1 and are the same as in ref. 33.

The *contingency model* infers the associations between the tone and the rotation direction to determine the prediction response. It relies on a three-level Hierarchical Gaussian Filter (HGF)[40,75] with low-, mid-, and high-level priors. Low-level priors correspond to the subjectively estimated chance of having a CW or CCW rotation (binomial), and to the conditional probability of a certain rotation direction given the tone. Mid-level priors capture the belief about the probabilistic association between the tone and the rotation direction. High-level priors model the stability of this association over the time course of the experiment (i.e., this association might remain stable or reverse).

The *contingency model* is coupled with a *perceptual model* which assesses whether the perceptual responses are influenced by associative learning (i.e., influence of the current hidden contingency), priming (i.e., influence of the preceding trial) and/or sensory memory (i.e., influence of the preceding ambiguous trial on the next ambiguous trial).

The behavioral data (prediction and perception responses) were fitted by eight models that included associative learning (A), priming (P) and/or sensory memory (S) in the perceptual model (i.e., none: 0, A, P, S, AP, AS, PS, APS). Model inversions were performed separately for each run. We used the HGF for binary inputs with the quasi-Newton Broyden-Fletcher-Goldfarb-Shanno minimization as optimization algorithm, implemented in the HGF 4.0 toolbox (TAPAS toolbox – Translational Algorithm for Psychiatry-Advancing Science, translationalneuromodeling.org/tapas/) in Matlab (R2020b version). To identify which of the eight models best fitted the data in each group, we used random-effect Bayesian Model Selection (BMS)[76] in SPM12 (http://www.fil.ion.ucl.ac.uk/spm/). This model comparison relies on the log-model evidence, which is calculated as the negative variational free energy. The model parameters were estimated using Bayesian Model Averaging (BMA) and compared between groups using Student's $t$ tests. As an additional way to check the model fit, behavioral data (percentages of correct predictions) were correlated with the estimated model parameters using Pearson correlation tests. Model and parameter recoveries are presented in Supplementary Note S2.

### fMRI analyses
The fMRI data were preprocessed and analyzed using Matlab 2020b and Statistical Parametric Mapping (SPM), version 12.

**fMRI preprocessing.** Prior to the preprocessing the anatomical image was manually reoriented to be centered on the middle anterior commissure. The preprocessing consisted of realigning the functional volumes of the five runs and of the localizer, slice timing correction (on timings, with the middle slice as reference), coregistration of the anatomical scan, segmentation, normalization (based on the standard template of the Montreal Neurological Institute, MNI) and smoothing with a Gaussian kernel of 6 mm (full width at half maximum). The Artifact detection toolbox (ART: art-2015-10 release, https://www.nitrc.org/projects/artifact_detect/) with a motion threshold of 2 mm was used to detect the outliers that were added as unique regressors to discard the identified outlier scans. On average, 0.5% and 1.5% of scans were excluded in the NT and ASD groups, respectively. Runs with more than 15% of outliers were excluded, resulting on the exclusion of the fourth and fifth runs of one ASD participant (i.e., all participants had five runs included, except for one ASD participant who had three runs included).

Finally, we conducted a denoising step using the CONN functional connectivity toolbox v17 (http://www.nitrc.org/projects/conn). Using

aCompCor[77], a principal component analysis on the cerebrospinal fluid and white matter masks was performed to remove physiological confounds from the signal.

**Whole-brain analysis.** In order to identify the neural correlates of predictions and prediction errors in individuals with and without ASD, we conducted a model-based fMRI analysis. We defined a General Linear Model (GLM) using two regressors (tone and rotation) and nine parametric modulators that were model quantities from the inverted behavioral model. These parametric modulators mostly corresponded to the subject-specific trajectories of prediction errors, prior mean, and precision.

The regressor *tone* modeled an event starting at the onset of the tone and was parametrically modulated by five model parameters: 3rd level prior precision $\widehat{\pi_3}$ and mean $|\widehat{\mu_3}|$, 2nd level prior precision $\widehat{\pi_2}$ and mean $|\widehat{\mu_2}|$, and low-level predictions $\mu_a$ (i.e., inferred conditional probability of a CW rotation). The 3rd level represents the belief about the volatility of the environment, while the 2nd level corresponds to the belief about the main contingency. The absolute prediction mean (e.g., $|\widehat{\mu_2}|$) models the strength of the belief (e.g., an individual with a high or low estimated $|\widehat{\mu_2}|$ believes that the tone is highly or not predictive of the rotation direction, respectively). The prediction precision denotes how quickly these estimates change across time. The regressor *rotation* modeled an event starting 600 ms after the presentation of the two vertical dots, i.e., at the onset of the rotation in unambiguous trials, and at the onset of the presentation of the two horizontal dots in ambiguous trials. The regressor *rotation* was parametrically modulated by four model parameters: 3rd level precision-weighted prediction errors $|\varepsilon_3|$, 2nd level precision-weighted prediction errors $|\varepsilon_2|$, perceptual prediction error $|\delta_q|$ (i.e., lower-level prediction error defined as $\delta_q = P(\Theta_1) - y_{perception}$) and the posterior probability of perceiving a CW rotation $P(\Theta_1)$. All regressors were convolved with a canonical hemodynamic response function. Note that the definition of the GLM was based on the study on NT[33], but third-level model quantities were added to the GLM.

In addition, the potential confounds were modeled as separate regressors of the GLM matrix and consisted of the six motion parameters, the ART-based outliers (if any) and the 10 first principal components identified with aCompCor in the denoising step.

At the first level, contrast images were computed for each regressor or parametric modulator at the individual level using t-statistics. At the second level, the mean of the contrasts across participants was compared to zero using a one-sample Student's *t* test. Activation patterns were compared between the two groups using independent two-sample t tests. When a significant group difference was found, we used Marsbar (release 0.44)[78] to extract the mean contrast estimate in the cluster ($p < 0.001$ at the voxel level, $p < 0.05$ at cluster level) and plot the distribution of contrast estimates across participants.

Voxel-based level thresholding was set at $p < 0.001$, and cluster-level extent thresholding was set at $p < 0.05$ (cluster-extent thresholds are estimated by Gaussian Random Field method implemented in SPM12). In the Results section, we report whether these results remain significant after Family Wise Error (FWE) correction at cluster level. The SPM toolbox bspmview (version 20161108)[79] was used to illustrate the brain activation patterns in Figs. 3, 4 and S1.

To assess if the perceptual bias induced in ambiguous trials generated activity in V5/MT, we conducted additional fMRI analyses, presented in Supplementary Note S3 and Supplementary Figure S2.

**Localizer-based analysis.** In addition to the whole-brain analysis, we conducted a region of interest (ROI)-based analysis, after having defined the ROIs using the localizer and the main task. The aim was to investigate whether the low-level prediction ($\mu_a$ or $1 - \mu_a$) about the rotation direction of the two dots (CW or CCW) would trigger activation in the retinotopic representation of the CW or CCW trajectories at the onset of the tone (i.e., whether the brain was proactive). The three-step procedure for the selection of the ROIs was the same as in the study by Weilnhammer and colleagues[33].

First, we specified a GLM with two box-car regressors in the localizer run: a *CW regressor* corresponding to the presentation of checkerboards over the upper-right and lower-left trajectories, and a *CCW regressor* corresponding to the checkerboard presentation over the lower-right and upper-right quadrants. Each of these regressors started at the onset of appearance of the checkerboard and lasted for 15 s. At the subject-level, we used the contrasts CW > CCW and CCW > CW, thresholded at $p < 0.05$ to identify the clusters responding to CW and CCW trajectories and used Marsbar to extract these clusters.

Second, in order to only select the voxels that were highly specific for the dot CW and CCW rotations in the main task, we constructed a second single-subject GLM. This GLM defined in the main task contained three regressors, all modeled as events (0 s duration): *CW regressor* at the onset of the CW rotation, *CCW regressor* at the onset of the CCW rotation and *Ambiguous regressor* at the onset of appearance of the dots in their horizontal position in ambiguous trials. At the subject-level, we used the contrasts CW > CCW and CCW > CW, thresholded at $p < 0.05$ (uncorrected) to identify the clusters responding to the CW and CCW rotations and used Marsbar to extract these clusters.

Third, we defined the CW and CCW ROIs by selecting the intercept of the masks defined in the two previous steps. At a single-subject level, these ROIs were used to mask the contrast $\mu_a$ (parametric modulator of the tone) in the GLM described in the "Whole-brain analysis". The individual contrast estimates were again extracted using Marsbar.

**DCM analysis.** We conducted a DCM[80] analysis to assess how mid-level and high-level priors and precision-weighted prediction errors influenced top-down and bottom-up connections, respectively, in each group. Using DCM has the particularity to make inferences about the causal relationships of activity patterns.

**Definition of the GLM.** The GLM used in the DCM analysis was the same as in the main analysis, but only included the regressors *Tone*, *Rotation*, $|\widehat{\mu_3}|$, $|\widehat{\mu_2}|$, $|\varepsilon_3|$ and $|\varepsilon_2|$.

**Definition of the regions of interest and extraction of the time series.** The DCM analysis involved the bilateral occipital cortex, left auditory cortex, the left OFC, and the left insula. The OFC has been identified as a key region encoding contextual priors[33,34,39,81,82]. We decided to include the insula among two other main candidates (i.e., the ACC and the caudate nucleus) as it was reported in articles investigating prediction errors across multiple levels of the hierarchy[31–33,41,83] and because a recent meta-analytical study[36] highlighted the role of the left insula in encoding prediction errors.

The regions of interest (ROIs) were identified using a two-step procedure: we first identified the peak of maximum activity across groups, and then at the individual level (same methods as in[84,85]). First, we defined the peak of activation across groups with $p < 0.001$ at voxel level and $p_{FWE-corr} < 0.05$ at cluster level. The contrast *Tone* vs. *baseline* was used to identify the peak of the auditory ROI (AUD: $x = -58$, $y = -14$, $z = 4$). Note that the maximum activity was in the left hemisphere and we decided to restrict to one region as the same auditory input was sent in each ear. We used the contrast *Rotation* vs. *baseline* masked with the CW or CCW mask from the localizer to get the peaks of the left occipital (OCC$_L$: $x = -32$, $y = -88$, $z = 4$) and right occipital (OCC$_R$: $x = 34$, $y = -84$, $z = 4$) ROIs. We used the contrast $|\widehat{\mu_2}|$ vs. *baseline* to select the peak coordinates of the left orbitofrontal cortex (OFC: $x = -26$, $y = 34$, $z = -10$). Using the contrast $|\varepsilon_2|$ vs. *baseline*, we extracted the peak activity in the left anterior insula (INS: $x = -36$, $y = 14$,

$z = -2$). As participants included in the DCM analysis needed to have similar numbers of runs, the ASD participant who had only 3 runs instead of 5 was excluded from this analysis (i.e., $n = 26$ NT, and $n = 24$ ASD in this analysis).

Then, we created a 12.5 mm-radius spheres around the maxima described above using Marsbar. Within this sphere, we extracted the coordinates of the peak activity of each participant using the same contrasts, with $p < 0.01$ for the OCC ROIs, and $p < 0.05$ for the other ROIs. For each contrast of interest, we then extracted the first eigenvector across all voxels that were above the threshold within a 6 mm-radius sphere centered on the individual peak coordinates.

**DCM specification and estimation, and model comparison.** The inputs were the sound (regressor *Tone*) entering through the AUD, and the visual input (regressor *Rotation*) entering through the $OCC_L$ and $OCC_R$.

We first specified a DCM with no modulatory influence to identify the intrinsic connections. All the models included bidirectional connections between the AUD and OFC. In addition, in $M1_{intrinsic}$, the OFC was connected to the OCC and INS, in $M2_{intrinsic}$, the OCC was connected to the INS and OFC, in $M3_{intrinsic}$, the INS was connected to the OCC and OFC, and in $M4_{intrinsic}$, these regions were all connected (i.e., OFC with the OCC and INS, and OCC connected with the INS). BMS showed that $M4_{intrinsic}$ was the best model in each group (protected exceedance probability in both NT and ASD: 1.00). So, we selected this model and added modulatory influences, as described below.

We considered modulatory influences of top-down connections from the OFC to the OCC and INS, and from the INS to the OCC by the prior mean at a high- and/or mid-levels, and modulatory influences of bottom-up connections by precision-weighted prediction errors at high- or mid-levels from the OCC to the OFC and INS, and from the INS to the OFC. We specified and estimated eight models to assess if the connections were modulated by priors and/or prediction errors at the mid and/or high levels of the hierarchy. The eight models were specified as follow: M1 without any modulation, M2 with a modulation by $|\widehat{\mu_2}|$, M3 with a modulation by $|\varepsilon_2|$, M4 with a modulation by $|\widehat{\mu_3}|$, M5 with a modulation by $|\varepsilon_3|$, M6 with modulations by $|\widehat{\mu_2}|$ and $|\varepsilon_2|$, M7 with modulations by $|\widehat{\mu_3}|$ and $|\varepsilon_3|$, and M8 with modulations by $|\widehat{\mu_2}|$, $|\varepsilon_2|$, $|\widehat{\mu_3}|$ and $|\varepsilon_3|$ (Fig. 5).

Finally, we used BMS to select the model that best fitted the fMRI data at the group level. We report both the protected exceedance probability and the model frequency for each model. We then used BMA to estimate the values of the posterior parameters of the DCM analyses (i.e., connections and modulation of connections).

### Reporting summary

Further information on research design is available in the Nature Portfolio Reporting Summary linked to this article.

## Data availability

The behavioral data generated in this study have been deposited in the Zenodo database under accession code (https://doi.org/10.5281/zenodo.7808070). The raw fMRI data are protected and are not available due to data privacy laws, but group-level fMRI results are available by request to the corresponding author. Source data are provided with this paper.

## Code availability

The codes of the behavioral and fMRI analyses of this study are available in a Zenodo deposit (https://doi.org/10.5281/zenodo.7808070).

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

## Acknowledgements

We would like to thank Naomi Couder and Joke Temmerman for their help in recruiting participants and collecting data. We also thank Ronald Peeters for his help in preparing the MRI exam card. Finally, we thank all the participants for their time. LAST was supported by a postdoctoral fellowship of the H2020 Marie Skłodowska-Curie Actions (PreCoASD). The research was financed by grants from a long-term structural funding from the Flemish Government (METH/14/02) to JW.

## Author contributions

All authors contributed to the conception of the study. L.A.S.T. and L.P. piloted the experiment and collected the data. L.A.S.T. and L.P. analyzed the behavioral data. L.A.S.T. analyzed the fMRI data. L.A.S.T., V.W., and P.S. modeled the data. J.W. was the main supervisor of the project. L.A.S.T. wrote the first draft of the article, and J.W. edited it. All authors approved the final version of the manuscript.

## Competing interests

The authors declare no competing interests.
