## [Peer Review File · Nature Communications]

Neural correlates of hierarchical predictive processes in autistic adultsREVIEWER COMMENTS

Reviewer #1 (Remarks to the Author):

Neural correlates of hierarchical predictive processes in autistic adults

The authors studied 26 NTs and 26 ASD using task fMRI to assess predictive processing during an audiovisual associative learning task. Behaviorally, there were no marked differences in predictive abilities between groups. Neurally, they observed activations for low level predictions in sensory areas while high level predictions involved transmodal areas in both groups. There were mainly differences between groups in prediction related activity for higher and mid-order predictions.

The authors interpret their findings to suggest overall intact neural mechanisms at the lower-level end of the cortical hierarchy in ASD, while neural correlates of predictions and predictions errors at higher levels appeared atypical. The paper is interesting, as it tries to model decision making under uncertainty in autism in a Bayesian context and as it relates ASD to atypical organization of neural hierarchies. I have a several comments and suggestions to further strengthen the authors' conclusions:

Introduction:

1) Can the authors provide further details on the inclusion criteria: were these individuals with 'idiopathic' ASD and no genetic (eg Fragile X) or metabolic diagnosis? Please provide further information on the medication taken by the individuals with ASD. Are the main findings consistent when individuals without ADHD, dyslexia, and Tourette are excluded?

Methods:

2) The authors convolve with their design matrix with a canonical hemodynamic response function, but their conclusions heavily invoke the notion of cortical hierarchy. Several prior studies have suggested spatially variable HRF properties across the cortex (eg <https://www.ncbi.nlm.nih.gov/pmc/articles/PMC5911213/pdf/nihms949811.pdf>), also differentiating higher and lower level areas. Moreover, there is prior work in non-human primates and in humans suggesting overall longer intrinsic neural time scales in higher order regions compared to sensory and motor regions (eg the work from Chaudhury et al in Neuron but also more recent fMRI studies in humans eg Ito and Cole 2020 Neuroimage). While I think its good they also perform an ROI based analysis in addition to their whole brain assessment, I would also recommend that the authors comment on this issue in the discussion. Moreover, they are encouraged to present a robustness analysis based on an alternative approach (eg FIR models?) that ensures that conclusions of the current work are not affected by potential variations in the HRF.

3) There appears some degree of selectivity for the choice of regions of interest for the DCM analysis. Aren't there recent extensions of the DCM framework that do in principle allow for whole brain modelling e.g. rDCM (<https://pubmed.ncbi.nlm.nih.gov/28259780/>)? If one wants to stick to a DCM based on a few set of nodes, one could also use prior data-driven maps of the putative cortical hierarchy to guide or at least support the ROI placement in this work. The principal gradient derived by Margulies et al (2016, PNAS) and/or microstructural differentiation reported by Paquola et al. (2018, PNAS) may come to mind, and/or classic depictions such as those by Mesulam et al (1998, Brain).

Results:

4) Can the authors clarify how they determined the cluster level $p < 0.05$ when findings were not based on FWE correction?

5) To follow up on point 3 above, it might be interesting to correlate findings in Figure 3 (e.g. uncorrected t-maps) with maps of functional/Margulies and microstructural/Paquola hierarchy, in order to assess associations to data-driven models of human cortical hierarchy organization in both groups. Ditto for Figure 4.

Reviewer #2 (Remarks to the Author):

A HGF model is used to assess how ASD and NT individuals encode predictions and prediction errors in the brain for motion discrimination. Importantly, the direction of the motion stimulus is contingent on an auditory cue presented prior to the motion stimulus. This AV contingency changes across trials.

There are a couple of issues:

1. In contrast to the author's previous study, the current study does not generate ambiguous stimuli (e.g. by removing stereodisparity cues). Instead, they treat static stimuli as ambiguous motion. It is not clear that static visual dots are perceived as ambiguous. Furthermore, the dual report paradigm may introduce additional complexities. For instance, Stocker et al. have shown that observers condition their response on their other responses for self-consistency in dual report paradigms. This may need to be explored or accounted for in the model.

2. The authors use a hierarchical HGF model. The model is not precisely described. Moreover, even after reading previous publications, I am left with questions about the model's assumptions (such as different priors for ambiguous and unambiguous stimuli) and update equations. Little justification for model choices is offered.

3. No model or parameter recovery is reported. This is even more important for this particular paper, because the authors do not find any significant differences in behavioural performance or model parameters across groups. Differences only arise for how model parameters are related to brain activations in NT and ASD individuals. Therefore, it is of utmost importance to ensure that parameters can be very precisely recovered. The HGF model includes a large number of parameters (number of parameters is not explicitly stated) and so it may not be that surprising that a subset of parameters predicts activations differently in NT and ASD.
4. The clarity of the paper could be improved. The introduction conflates interpretation and results. For instance, it remains controversial whether forward connections convey prediction errors during perceptual inference.
5. The authors suggest that contingencies influence observers' percept of static dots. Numerous studies have shown representations in MT/V1 associated with observers' percept. Perhaps the authors could perform additional decoding analyses to support their claims, i.e. show that AV contingencies influence the representation decoded from MT.
6. It is unclear why different DCMs are specified for priors and prediction errors. As both priors and prediction errors need to be computed on every trial, I would think both aspects should be integrated in one DCM.

Neural correlates of hierarchical predictive processes in autistic adults

--- Responses to the reviewers ---

We are grateful to the reviewers for their careful reading, their encouraging remarks and suggestions that helped us improve the manuscript. Please find below our answers to all questions. Modifications made to the paper appear in between quotation marks and in blue here and are also written in blue in the revised main document.

- **Reviewer #1 (Remarks to the Author):**

Neural correlates of hierarchical predictive processes in autistic adults

The authors studied 26 NTs and 26 ASD using task fMRI to assess predictive processing during an audiovisual associative learning task. Behaviorally, there were no marked differences in predictive abilities between groups. Neurally, they observed activations for low level predictions in sensory areas while high level predictions involved transmodal areas in both groups. There were mainly differences between groups in prediction related activity for higher and mid-order predictions.

The authors interpret their findings to suggest overall intact neural mechanisms at the lower-level end of the cortical hierarchy in ASD, while neural correlates of predictions and predictions errors at higher levels appeared atypical. The paper is interesting, as it tries to model decision making under uncertainty in autism in a Bayesian context and as it relates ASD to atypical organization of neural hierarchies. I have a several comments and suggestions to further strengthen the authors' conclusions:

Introduction:

1) Can the authors provide further details on the inclusion criteria: were these individuals with 'idiopathic' ASD and no genetic (eg Fragile X) or metabolic diagnosis? Please provide further information on the medication taken by the individuals with ASD. Are the main findings consistent when individuals without ADHD, dyslexia, and Tourette are excluded?

Answer: Indeed, all the autistic participants included in this study had an idiopathic form of ASD, as no genetic or metabolic causes had been identified. This is now specified in the Methods:

“Autistic participants received their diagnoses from a multidisciplinary Expertise Centre for Autism [...] and had idiopathic ASD.” (p. 6).

Moreover, we now provide the whole list of medications reported by the autistic participants:

“Eleven ASD participants reported taking one or several medications (Abilify (2), Asaflow (1), Bufonix (1), Celecoxib (1), Deanxit (1), Depakine (1), Escitalopram (1), Fluoxetine (1), Fluoxone

(1), Hydrea (1), L-Thyroxine (3), Medikinet (1), Melatonine (1), Montelukast (1), No tril en (1), Redomex (1), Ritalin (1), Trazadone (1), Venlafaxine (1), Welbutrin (1)).” (p. 6).

Regarding the participants who had comorbidities, first note that without these five participants, conclusions on the main demographics and questionnaires remain the same (mean age: 31.7 with $n = 21$ (vs. 32.2 with $n = 26$), mean AQ: 32.4 with $n = 21$ (vs. 32.1 with $n = 26$), mean IU: 43.0 with $n = 21$ (vs. 42.7 with $n = 26$), mean GSQ: 53.7 with $n = 21$ (vs. 53.3 with $n = 26$), mean IQ: 113.0 with $n = 21$ (vs. 112.1 with $n = 26$)).

The behavioral and modeling results did not change after excluding the five participants who had comorbidities:

- Percentage of correct prediction response above chance level in ASD (75% with $n = 21$ ASD, 73% with $n = 26$ ASD, $p < .0001$ with both the small and large samples), and no significant group difference on the percentage of correct prediction responses (76% in the NT group, group comparison: $p = .76$).
- Percentage of correct perception response in unambiguous trials not significantly different between groups (99% with $n = 21$ and $n = 26$ ASD, 99% in the NT group).
- Percentage of perception responses following the main contingency in ambiguous trials above chance level (67% with $n = 21$ ASD, 66% with $n = 26$ ASD, $p < .0001$), and not significantly different between groups (72% in the NT group, $p = .17$).
- No significant group difference on any of the model parameters (π_a : 1.33 with $n = 21$ ASD, 1.32 with $n = 26$ ASD, 1.37 in NT; ω_2 : -0.75 with $n = 21$ ASD, -0.89 with $n = 26$ ASD, -0.72 in NT; ω_3 : -6.25 with $n = 21$ and 26 ASD, -6.23 in NT).

The main findings on the fMRI analyses also did not change when excluding the autistic participants who had comorbidities. First, the global pattern of the activation maps did not change after excluding the five participants who had comorbidities.

Second, the main results of the localizer analysis also did not change after removing the five participants with comorbidities. Indeed, hearing a tone that was predictive of a CW or CCW rotation yielded activity in the CW or CCW mask, respectively, in 70% of the ASD participants (68% in CW mask and 72% in CCW mask with $n = 26$).

Finally, the contrast estimates extracted in the clusters showing group differences (Fig. 3 and 4) also revealed that the participants with comorbidities did not influence the results as their estimates were always within a one (or sometimes two) standard deviation(s) from the mean and never appeared to be outliers. After removing the five participants with comorbidities, the t-tests on contrast estimates remained significant in the six clusters where group differences were found (all p-values $< .0002$).

We have added a sentence in the Methods section to mention that participants did not appear to have influenced the behavioral nor fMRI analyses:

“These participants who had comorbidities are included in the analyses, but note that the results of the behavioural and fMRI analyses did not change after removing these five participants.” (p. 6).

Methods:

2) The authors convolve with their design matrix with a canonical hemodynamic response function, but their conclusions heavily invoke the notion of cortical hierarchy. Several prior studies have suggested spatially variable HRF properties across the cortex (eg <https://www.ncbi.nlm.nih.gov/pmc/articles/PMC5911213/pdf/nihms949811.pdf>), also differentiating higher and lower level areas. Moreover, there is prior work in non-human primates and in humans suggesting overall longer intrinsic neural time scales in higher order regions compared to sensory and motor regions (eg the work from Chaudhury et al in Neuron but also more recent fMRI studies in humans eg Ito and Cole 2020 Neuroimage). While I think its good they also perform an ROI based analysis in addition to their whole brain assessment, I would also recommend that the authors comment on this issue in the discussion. Moreover, they are encouraged to present a robustness analysis based on an alternative approach (eg FIR models?) that ensures that conclusions of the current work are not affected by potential variations in the HRF.

Answer: We thank the reviewer for the references and appreciate their concern. However, we would like to highlight that the design of our study is not the most optimal to perform FIR analyses. Indeed, conducting FIR analyses can be quite noisy and therefore less powerful. In the study by Ito and Cole, 2020, the authors were indeed able to use a FIR model to process their data, but it should be noted that they had a block design with a TR of 720 ms and a multiband factor of 8, whereas we have an even-related design with a TR of 2 s and a multiband factor of 2.

Even though our design is clearly not optimized to run FIR analyses, we conducted a simple FIR analysis on the two regressors of our main GLM (i.e., Tone and Rotation), by defining 10 time points every 2 s. We specified a FIR model and conducted first and second-level analyses. Based on the main GLM presented in the manuscript, we selected the coordinates of the four clusters associated with the highest T values when ASD and NT groups were pooled together, for the contrast *Rotation* vs. baseline (i.e., right occipital cortex, left putamen, right superior frontal gyrus, right middle frontal gyrus), and *Tone* vs. baseline (i.e., left Heschl gyrus, right Heschl gyrus, left precentral gyrus and right superior frontal gyrus). Using Marsbar, we created 10 mm-radius spheres around these coordinates. Then, using Marsbar, we extracted the contrast estimates in these ROIs (i.e., 4 regions per regressor) across the time points modeled with FIR. We focused

on the four first time points, and conducted a group (NT or ASD) by ROIs (4 regions) ANOVA at each of these time points. On the four regions tested for the regressor *Rotation*, there were no group by ROI interactions (F values ranging from 0.85 to 0.96, $p > .41$). For the regressor *Tone*, there were no group by ROI interaction either (F values ranging from 0.34 to 1.50, $p > .22$). As there was no group by ROI interaction, we conclude that our fMRI results were not affected by different temporal patterns of the HRF between groups, but we insist that any conclusion from this FIR analysis must be considered cautiously as our design is not well suited for this kind of analyses.

As advised by the reviewer, we discuss this point in the Discussion and added the references mentioned by the reviewer:

“Using a design with shorter TRs could have revealed variability across different regions of the cortex in the temporal shape of the HRF⁸², with faster responses in sensory regions than in associative areas⁸³.” (p. 28).

3) There appears some degree of selectivity for the choice of regions of interest for the DCM analysis. Aren't there recent extensions of the DCM framework that do in principle allow for whole brain modelling e.g. rDCM (<https://pubmed.ncbi.nlm.nih.gov/28259780/>)? If one wants to stick to a DCM based on a few set of nodes, one could also use prior data-driven maps of the putative cortical hierarchy to guide or at least support the ROI placement in this work. The principal gradient derived by Margulies et al (2016, PNAS) and/or microstructural differentiation reported by Paquola et al. (2018, PNAS) may come to mind, and/or classic depictions such as those by Mesulam et al (1998, Brain).

Answer: The simulations run in the article by Frassle et al., 2017, suggest that the current formulation of rDCM would not give reliable results given our acquisition parameters. Indeed, it is specified that there are high requirements in terms of data quality, with fast TR and high SNRs, to be able to use rDCM. We therefore believe that our experimental design is not suited to perform rDCM.

However, following the 6th comment of Reviewer 2, we present a new version of the DCM results (see response below).

Results:

4) Can the authors clarify how they determined the cluster level $p < 0.05$ when findings were not based on FWE correction?

Answer: We were referring to cluster-level extent thresholding (in contrast to peak-level thresholds), as implemented in SPM, where the cluster is measured in units of contiguous voxels. The cluster-extent based thresholding method relies on Gaussian Random Field methods, implemented in SPM12. We clarified it in the Methods:

“[...] cluster-level extent thresholding was set at $p < .05$ (cluster-extent thresholds are estimated by Gaussian Random Field method implemented in SPM12).” (p. 13).

5) To follow up on point 3 above, it might be interesting to correlate findings in Figure 3 (e.g. uncorrected t-maps) with maps of functional/Margulies and microstructural/Paquola hierarchy, in order to assess associations to data-driven models of human cortical hierarchy organization in both groups. Ditto for Figure 4.

Answer: We thank the reviewer for raising this interesting comment on the similarities between the hierarchy identified in our study and the functional and microstructural gradients. Results about the neural correlates of mid and high-level predictions indeed resemble the principal gradient described by Margulies et al. 2016, while the neural correlates of low-level predictions resemble their second gradient. Recent findings on the microstructural gradients also brought interesting ideas for the Discussion. We now comment on the relationship between our fMRI findings and their maps in the Discussion (pp. 25-26):

“The regions identified along this hierarchy resemble the principal gradient of macroscale organization in the human connectome⁶⁹, as higher-level predictions involve regions belonging to the principal gradient, i.e., transmodal regions including the default mode network and fronto-parietal regions, while low-level predictions resemble the map of the second functional gradient. Regions of the default mode network are at the upper end of the topographical hierarchy and integrate information from several sensory modalities to get abstract representations⁶⁹. This functional gradient from primary sensory regions to transmodal regions is therefore in line with the hierarchical neural correlates of predictions that we identified. Interestingly, a recent study investigated the microstructural profiles of these regions⁷⁰. They identified gradual cytoarchitectural variations that globally followed the functional gradient, but the transmodal cortices had a less hierarchical organization⁷⁰. The rich and less-constrained interconnectivity of these transmodal regions would be central to underlie flexible cognitive functions⁷⁰, which may allow representing higher-level abstract representations, for instance about the stability of predictions over time.”

Reviewer #2 (Remarks to the Author):

A HGF model is used to assess how ASD and NT individuals encode predictions and prediction errors in the brain for motion discrimination. Importantly, the direction of the motion stimulus is contingent on an auditory cue presented prior to the motion stimulus. This AV contingency changes across trials. There are a couple of issues:

1) In contrast to the author's previous study, the current study does not generate ambiguous stimuli (e.g. by removing stereodisparity cues). Instead, they treat static stimuli as ambiguous motion. It is not clear that static visual dots are perceived as ambiguous. Furthermore, the dual report paradigm may introduce additional complexities. For instance, Stocker et al. have shown that observers condition their response on their other responses for self-consistency in dual report paradigms. This may need to be explored or accounted for in the model.

Answer: The paradigm is exactly the same as in Weinhhammer et al., 2018, which also did not include any stereodisparity of the cues. As in the original study by Weinhhammer et al., ambiguous trials indeed consist of two dots which appear in the vertical and then horizontal position, with spatiotemporal properties that give rise to apparent motion which can be clockwise (CW) or counterclockwise (CCW). In the unambiguous trials instead, the dots are rotating physically rotating within 33 ms, which is sufficient to perceive an actual motion in the CW or CCW rotation.

First, the behavioral responses in ambiguous trials show that the percentage of trials perceived according to the current contingency is above chance level in both groups (72% in the NT group and 66% in the ASD group, $p < .0001$ in both groups). This means that participants indeed had the impression that the two dots were rotating (otherwise, such percentages would not differ from chance level). If we consider individual data, these percentages were below 50% in only 3 out of 52 participants.

Furthermore, we have now added results from the confidence rating task which took place outside of the MRI, after completing the main experiment and. Initially, these data were not included in the manuscript to focus on the fMRI results and to avoid a too long description of the behavioral results, as it served as quality checks. Data showed that ambiguous trials were indeed perceived as more uncertain than unambiguous trials. We added information regarding this task in the Methods (p. 9):

“After finishing the fMRI experiment, participants completed a short computer task as in³³ to assess the perceptual quality of the ambiguous trials. The structure of this task was the same as the main task, but trials included a third response screen showing the options “1. Very sure”, “2. Quite sure”, “3. Quite unsure”, “4. Very unsure” (displayed for 2600 ms). Participants used the numbers 1 to 4 to indicate how confident they were about their perception response. There were 48 trials, including 50% of ambiguous trials and 50% of unambiguous trials. To calculate a mean confidence rating, the 1 (Very sure) to 4 (Very unsure) scale was transformed into a 100% to 0% certainty scale.”

And in the Results (p. 17):

“Finally, in the confidence rating task, a repeated-measure ANOVA on confidence rating scores showed a main effect of ambiguity ($F(1,50) = 40.0, p < .0001$), with ambiguous trials being rated as more uncertain than unambiguous trials ($60\% \pm 30$ certain vs. $90\% \pm 14$ certain, $t(51) = 6.4, p < .0001, d = 0.88$), but revealed no group effect nor interaction.”

When debriefing about the experiment with the participants after the confidence rating task, we asked them whether they had the impression that the dots were not turning. Only two NT participants and five ASD participants reported that they sometimes had the impression that the dots were not turning. But note that we asked this question after the participant completed the confidence rating task, which might have biased their judgment as this task made them wonder about the certainty of their percept. Yet, none of the participants reported spontaneously during or after the experiment that they noticed that the dots were sometimes not turning.

Given our behavioral results from the main task and from the confidence rating task, as well as the feedback from the participants and the results from Weirhammer et al., 2018, we believe that we succeeded at generating ambiguous percepts.

Regarding the second part of this comment, using dual report paradigm may indeed introduce additional complexities, such as introducing a kind of repetition bias to show more consistency. We have added it as a limitation in the Discussion (p. 28):

“We chose to rely on a dual-report paradigm with both prediction and perception responses to get an explicit measure of prediction learning (where tone could not be simply ignored) and to get an implicit measure of perceptual bias, but using such paradigm might bias responses as participants often tend to ensure self-consistency⁸¹.”

In the associative learning task by Lawson et al. (2017), participants only gave a perception response (but no prediction response). They assessed how priors were learned based on response times and error rates, and showed that these variables were not modulated by the predictability of the stimuli in autistic participants, in contrast with NT. The authors concluded that autistic adults responded inflexibly to expected or unexpected outputs, and had overestimated the volatility of their environment. However, autistic participants might have simply ignored the tones, which were irrelevant to perform the task and whose underlying association with visual outputs was complex and changing. In our task, using a dual-report paradigm was a strength to ensure that tones were not ignored and that participants really tried to learn the cue-outcome association.

2) The authors use a hierarchical HGF model. The model is not precisely described. Moreover, even after reading previous publications, I am left with questions about the model’s assumptions (such as different priors for ambiguous and unambiguous stimuli) and update equations. Little justification for model choices is offered.

Answer: We have now added the full mathematical model description as supplementary material (pp. 10 to 15 in the supplementary information file) to explain how the HGF model is implemented, and we also provide additional information. We rely on the same equations and model assumptions as in Weilhhammer et al., 2018. The model structure and list of model parameters are presented in Figure 2A and 2B. We hope that adding the mathematical description has clarified the model.

3) No model or parameter recovery is reported. This is even more important for this particular paper, because the authors do not find any significant differences in behavioural performance or model parameters across groups. Differences only arise for how model parameters are related to brain activations in NT and ASD individuals. Therefore, it is of utmost importance to ensure that parameters can be very precisely recovered. The HGF model includes a large number of parameters (number of parameters is not explicitly stated) and so it may not be that surprising that a subset of parameters predicts activations differently in NT and ASD.

Answer: Indeed, this is an important point, and we have now performed simulations and report model and parameter recovery in the Supplementary Material. This additional analysis showed that both the models and parameters could be recovered. We refer to this analysis in the Methods section (p. 11):

“Model and parameter recoveries are presented as Supplementary Material, Appendix S2.”

Furthermore, the details of these simulations are presented in the Supplementary Material (p. 16):

“Appendix S2: Model and parameter recoveries

To assess the validity of our modeling approach, both in terms of discriminability between models and parameter recovery, we performed simulations.

First, using Bayesian Model Selection comparing the eight models on the simulated data, we observed that model M5 best explained the data in both groups (protected exceedance probabilities: 1.00). This is in line with our findings, reported in Figure 2.

Then, we estimated the simulated parameters using Bayesian Model Averaging. As expected, the simulated parameters were correlated strongly positively with the estimated parameters (tested using Pearson correlations). Indeed, the estimated and simulated associative learning precision π_a parameters were correlated (NT: $r = .93$, $p < .001$; ASD: $r = .94$, $p < .001$). In addition, the estimated and simulated second and third-level learning rates were correlated in the NT group (ω_2 : $r = .91$, $p < .001$, ω_3 : $r = .67$, $p < .001$) and in the ASD group (ω_2 : $r = .96$, $p < .001$, ω_3 : $r = .78$, $p < .001$).”

Finally, we would like to note that the model best fitting the data in both groups is not the model with the largest number of parameters among the eight models that we tested, which suggests a

decreased risk of overfitting. Indeed, the winning model is the one only including associative learning but no priming or sensory memory parameters.

4) The clarity of the paper could be improved. The introduction conflates interpretation and results. For instance, it remains controversial whether forward connections convey prediction errors during perceptual inference.

Answer: We apologize for the lack of clarity, and we have worked on the manuscript to improve its overall clarity, and in particular on the Introduction to better distinguish interpretations from results. We also specified that the prediction errors are simply hypothesized to be conveyed through forward connections.

5) The authors suggest that contingencies influence observers' percept of static dots. Numerous studies have shown representations in MT/V1 associated with observers' percept. Perhaps the authors could perform additional decoding analyses to support their claims, i.e. show that AV contingencies influence the representation decoded from MT.

Answer: Indeed, it would be relevant to show such activation in MT for ambiguous trials, to assess whether they are perceived as rotating dots. However, the ambiguous trials with static dots only represent 12.5% of the trials (i.e., 9 trials per run), which is too little to be able to run decoding analyses.

Interestingly, in a recent fMRI study, Haarsma and colleagues (2022, BioRxiv) used an associative learning paradigm where a tone was followed by a grating with a certain orientation or simply by noisy patches. They were interested in investigating whether a false percept could be observed, i.e., whether the expected orientation would be perceived in noisy patches when the tone was predictive of that orientation. They were only able to observe it in the middle input layer of V2, and insisted on the fact that this effect could only be observed thanks to the use of a 7T MRI scanner.

However, it should be noted that previous studies successively managed to decode the perceived rotation direction of dots from ambiguous stimuli (e.g., Schmack et al., 2017).

Based on the reviewer's comment, we added this point as a perspective in the Discussion (p. 26):

“Future studies including more ambiguous trials could perform decoding analyses (e.g., such as in ⁷¹) to assess if the perceptual bias induced in ambiguous trials generates activity in motion area MT.”

As we could not perform decoding analyses on the restricted number of ambiguous trials that we had, we specified another GLM to model ambiguous trials and used an anatomical mask of

MT/V5 to assess if ambiguous trials were associated with activity in this region. These additional analyses were added in the Supplementary Material, Appendix 3:

“Appendix S3: Activation by ambiguous trials in MT/V5

In order to determine if the presentation of ambiguous trials generated activity in MT/V5, we specified another GLM with Unambiguous rotation and Ambiguous rotation as regressors. These regressors were coded as events starting at the appearance of the two vertical dots. To account for additional variance, we also included the following regressors: Tone (starting at the beginning of the tone, and lasting for 500 ms), Prediction response (starting at the onset of the participant’s prediction response, and coded as an event), Perception response (starting at the onset of the participant’s perception response, and coded as an event). In addition, as in the other analyses, the potential confounds were modeled as separate regressors of the GLM matrix and consisted of the six motion parameters, the ART-based outliers (if any) and the 10 first principal components identified with aCompCor in the denoising step. At the first level, contrast images were computed for each regressor at the individual level using t-statistics. At the second level, the mean of the contrasts across participants was compared to zero using a one-sample Student's t-test.

We used the Anatomy Toolbox (SPM12) to create a structural mask encompassing the left and right MT/V5 areas. At the group level, the contrast Ambiguous rotation vs. baseline were masked with this MT/V5 mask, and voxel-based level thresholding was set at $p < .001$, while cluster-level extent thresholding was set at $p < .05$.

We found significant activation in the MT/V5 mask in response to ambiguous trials both in the NT group (left cluster: $x = -44, y = -70, z = 2, T = 8.7, p < .01, p_{FWE-corr} = .06$; right cluster: $x = 48, y = -66, z = -2, T = 6.8, p < .01, p_{FWE-corr} = .17$) and in the ASD group (left cluster: $x = -42, y = -68, z = 2, T = 10.1, p < .01, p_{FWE-corr} = .06$; right cluster: $x = 50, y = -68, z = 2, T = 7.9, p < .01, p_{FWE-corr} = .17$). Results are illustrated below.”

6) It is unclear why different DCMs are specified for priors and prediction errors. As both priors and prediction errors need to be computed on every trial, I would think both aspects should be integrated in one DCM.

Answer: Indeed, initially we had chosen to perform two DCMs to have a smaller model space and focus on either priors or prediction errors. However, following the reviewer's comments, we have run new DCM analyses which combine both priors and prediction errors, and integrate a larger set of regions, i.e., the bilateral occipital cortices, the left auditory cortex, insula and orbitofrontal cortex.

We adapted the Methods accordingly (pp. 14-16):

“The GLM used in the DCM analysis was the same as in the main analysis, but only included the regressors Tone, Rotation, $|\widehat{\mu}_3|$, $|\widehat{\mu}_2|$, $|\varepsilon_3|$ and $|\varepsilon_2|$.

The DCM analysis involved the bilateral occipital cortex, left auditory cortex, the left OFC and the left insula. The OFC has been identified as a key region encoding contextual priors^{33,34,39,62,63}. We decided to include the insula among two other main candidates (the ACC and the caudate nucleus) as it was reported in articles investigating prediction errors across multiple levels of the hierarchy^{31-33,41,64} and because a recent meta-analytical study³⁶ highlighted the role of the left insula in encoding prediction errors.

[...]

The inputs were the sound (regressor Tone) entering through the AUD, and the visual input (regressor Rotation) entering through the OCC_L and OCC_R.

We first specified a DCM with no modulatory influence to identify the intrinsic connections. All the models included bidirectional connections between the AUD and OFC. In addition, in $M1_{intrinsic}$, the OFC was connected to the OCC and INS, in $M2_{intrinsic}$, the OCC was connected to the INS and OFC, in $M3_{intrinsic}$, the INS was connected to the OCC and OFC, and in $M4_{intrinsic}$, these regions were all connected (i.e., OFC with the OCC and INS, and OCC connected with the INS). BMS showed that $M4_{intrinsic}$ was the best model in each group (protected exceedance probability in both NT and ASD: 1.00). So, we selected this model and added modulatory influences, as described below.

We considered modulatory influences of top-down connections from the OFC to the OCC and INS, and from the INS to the OCC by the prior mean at a high- and/or mid-levels, and modulatory influences of bottom-up connections by precision-weighted prediction errors at high- or mid-levels from the OCC to the OFC and INS, and from the INS to the OFC. We specified and estimated eight models to assess if the connections were modulated by priors and/or prediction

errors at the mid and/or high levels of the hierarchy. The eight models were specified as follows: M1 without any modulation, M2 with a modulation by $|\hat{\mu}_2|$, M3 with a modulation by $|\varepsilon_2|$, M4 with a modulation by $|\hat{\mu}_3|$, M5 with a modulation by $|\varepsilon_3|$, M6 with modulations by $|\hat{\mu}_2|$ and $|\varepsilon_2|$, M7 with modulations by $|\hat{\mu}_3|$ and $|\varepsilon_3|$, and M8 with modulations by $|\hat{\mu}_2|$, $|\varepsilon_2|$, $|\hat{\mu}_3|$ and $|\varepsilon_3|$ (Figure 5).”

We also update the Results (p. 23):

“We assessed whether top-down and bottom-up connections were modulated by high-level and/or mid-level predictions and/or precision-weighted prediction errors, respectively (Figure 5.A). The BMS (Figure 5.B) revealed that M8 was the best model in the NT group (protected exceedance probability: 0.95, model frequency: 0.41). In the ASD group, the results were less clear, with M8 being the best model (protected exceedance probability: 0.54, model frequency: 0.28), followed by M6 (protected exceedance probability: 0.33, model frequency: 0.25).

A BMA on the posterior parameters (Figure 5.C) showed that all the intrinsic connections were significantly different from zero (p -values $< .001$ in both groups), except for the connection from the INS to the OFC (NT: $p = .10$, ASD: $p = .52$). In NT, there were modulatory effects by $|\varepsilon_3|$ on the connections from the OCC_L and OCC_R to the INS (p -values < 0.005), and by $|\varepsilon_2|$ on the connections from the OCC_R to the INS (p -values < 0.05). In NT, there was a modulatory effect by $|\hat{\mu}_2|$ on connections from the OFC to the OCC_L ($p < .01$), and non-significant trends for modulations by $|\hat{\mu}_3|$ on connections from the OFC to the OCC_L ($p = .067$) $|\hat{\mu}_2|$ on connections from the INS to the OCC_L ($p = .099$). In ASD, there were modulatory effects by $|\varepsilon_3|$ on the connections from the OCC_L to the INS ($p < 0.05$) and to the OFC ($p < 0.05$). In ASD, there was a modulatory effect by $|\hat{\mu}_3|$ on connections from the OFC to the INS ($p < .005$), and a non-significant trend toward a modulation by $|\hat{\mu}_2|$ on this connection ($p = .062$). Between-group comparison only showed a non-significant trend toward a group difference on the modulation by $|\varepsilon_2|$ on the connection from the OCC_L to the INS ($t(48) = 1.7$, $p = 0.89$, NT: $0.02 \text{ Hz} \pm 0.18$, ASD: $-0.16 \text{ Hz} \pm 0.48$).”

Finally, we made small adjustments in the Discussion, but the main conclusions did not change. Figure 5 was also adjusted to present the new models and results.

REVIEWER COMMENTS

Reviewer #1 (Remarks to the Author):

I would like to thank the authors for their thoughtful revisions, and do now recommend the paper for publication.

Reviewer #2 (Remarks to the Author):

Thanks for responding so thoroughly to previous comments. The key aspect that requires further attention is parameter recovery. The revised manuscript reports correlations between true and recovered parameters. There are two issues:

1. More methodological details are needed to allow assessment of the parameter recovery (e.g. which parameter range, how many simulations etc.)
 2. Because the key finding is a difference in parameter estimates across groups, the following is in addition required:
 - simulate behavioural responses for each of your ASD and NT individuals using the parameter estimates you obtained from the initial fit of the winning model.
 - fit the same model to these simulated responses for each simulated ASD and NT individual
 - perform the same statistics as for your original data sets
- repeat steps 1-3 multiple times and report the fraction of simulations in which the difference across groups is significant as an index for statistical power.

see e.g. Wilson and Collins (2019), eLIFE

Neural correlates of hierarchical predictive processes in autistic adults

--- Responses to the reviewers ---

We are grateful to the reviewers for their suggestions that helped us improve the manuscript. Please find below our answers to the remaining question. We are sorry that it took three months to send this response, the first author is currently in maternity leave. Modifications made to the paper appear in blue here and in the revised manuscript.

- **Reviewer #1 (Remarks to the Author):**

I would like to thank the authors for their thoughtful revisions, and do now recommend the paper for publication.

Answer: We thank the reviewer for their feedback.

- **Reviewer #2 (Remarks to the Author):**

Thanks for responding so thoroughly to previous comments.

The key aspect that requires further attention is parameter recovery. The revised manuscript reports correlations between true and recovered parameters. There are two issues:

More methodological details are needed to allow assessment of the parameter recovery (e.g. which parameter range, how many simulations etc.)

Because the key finding is a difference in parameter estimates across groups, the following is in addition required:

- simulate behavioural responses for each of your ASD and NT individuals using the parameter estimates you obtained from the initial fit of the winning model.
- fit the same model to these simulated responses for each simulated ASD and NT individual
- perform the same statistics as for your original data sets
- repeat steps 1-3 multiple times and report the fraction of simulations in which the difference across groups is significant as an index for statistical power. see e.g. Wilson and Collins (2019), eLIFE

Answer: We thank the reviewer for giving more precise information about the requirements for the simulations. In line with the results reported in the main document, the simulated parameters did not differ between groups. The details of these simulations are given as Supplementary Material (p. 17) and copied below:

“Appendix S2: Model and parameter recoveries

Simulations were performed to assess the validity of our modeling approach, both in terms of discriminability between models and parameter recovery. More specifically, the behavioral responses of each of the 52 participants were simulated 100 times, using the parameter estimates obtained from the initial fit of the winning model (Associative learning model). Priors ranged from 0.18 to 2.88 for π_a , from -5.26 to 0.41 for ω_2 and from -6.66 to -5.94 for ω_3 . For each participant, the simulated behavioral responses were fitted by eight models (i.e., none: 0, A, P, S, AP, AS, PS, APS). Model inversions were performed separately for each run.

First, using Bayesian Model Selection comparing the eight models on the simulated data, we observed that model the Associative learning model best explained the data in both groups (protected exceedance probabilities: 1.00) in all the simulations, which is in line with our findings reported in Figure 2. For each participant, the simulated parameters of the Associative learning model were averaged over the five runs. As expected, the simulated parameters were correlated positively with the estimated parameters (tested using Pearson correlations). Indeed, the estimated and simulated associative learning precision π_a parameters were significantly correlated in 100% of the simulations in the NT and ASD groups (mean $r = .88$ in NT, $r = .87$ in ASD). The estimated and simulated second and third-level learning rates were also significantly correlated in 100% of the simulations in the NT and ASD groups (mean $r = .91$ for ω_2 and $r = .68$ for ω_3 in NT, mean $r = .95$ for ω_2 and $r = .77$ for ω_3 in ASD).

Finally, we assessed whether the simulated parameters differed between groups, using two-sample t-tests in each of the 100 simulations. In line with our findings reported in the Results section, the simulated π_a , ω_2 and ω_3 did not differ significantly between groups. Indeed, for each of the simulated parameters, 0% of the simulations showed significant group differences (π_a : t values ranging from -0.42 to 1.88, ω_2 : t values ranging from -0.18 to 1.49, ω_3 : t values ranging from -1.64 to 1.20).”

REVIEWERS' COMMENTS

Reviewer #2 (Remarks to the Author):

Thanks for addressing my comments. Congratulations to this nice paper!